# Mitigating Reward Hacking in LLM-based Recommendation: A Preference Optimization Approach

Heyu Chen [* 1]   Junkang Wu [1]   Guoqing Hu [1]   Kexin Huang [1]   Xiang Wang [1]   Jiancan Wu [1 2]

## Abstract

Post-training adaptation has become the central paradigm for leveraging large language models (LLMs) in recommendation. While recent preference optimization methods, such as Direct Preference Optimization (DPO), enhance pairwise preference discrimination, they remain vulnerable to *reward hacking*: models exploit imperfections in reward signals, leading to inflated training metrics without genuine recommendation gains. We analyze this issue from a gradient perspective and formalize the concept of the $\varepsilon$-*insensitive region*, where pairwise updates exert little influence on the ordering between positives and unsampled negatives. Under the Bradley–Terry model, we further show that these regions can occupy a substantial fraction of the preference space, inevitably leading to misaligned rankings. To address this issue, we propose Simulated Preference Optimization for Reward-hacking mitigation using Pseudo-negatives (SIRIUS). Our framework introduces pseudo-negative samples to enrich contrastive signals and reduce the prevalence of $\varepsilon$-insensitive regions. Extensive experiments on three public benchmarks show that SIRIUS consistently improves ranking quality and effectively mitigates reward hacking, providing both theoretical and practical insights for advancing LLM-based recommendation. Our code is available at https://github.com/heyucchen/sirius

## 1. Introduction

Post-training adaptation has become the dominant paradigm for modeling user preferences in LLM-based recommenda-tion. Early methods—prompting (Gao et al., 2023b; Geng et al., 2022; Dai et al., 2023) and supervised fine-tuning (SFT) (Bao et al., 2023b; Liao et al., 2024b; Bao et al., 2023a; Lin et al., 2024; Zhang et al., 2023)—cast user histories and item descriptions as text for next-token prediction; despite strong results, they lack explicit negative modeling, limiting fine-grained preference contrasts. Preference optimization addresses this with pairwise objectives: Direct Preference Optimization (DPO) (Rafailov et al., 2023), a lightweight alternative to RLHF (Christiano et al., 2017), is analogous to Bayesian Personalized Ranking (BPR) (Rendle et al., 2012), and recent variants tailored to LLM recommenders report improved discrimination and ranking (Chen et al., 2024b; Liao et al., 2024a; Gao et al., 2025).

However, preference optimization is vulnerable to **reward hacking**: models exploit imperfections in reward signals to inflate training metrics without genuinely improving alignment (Amodei et al., 2016). While related effects are documented in LLM alignment (Rafailov et al., 2023; Rashidinejad & Tian, 2024), their mechanisms in recommendation remain underexplored. In LLM-based recommendation (Figure 1), reward hacking appears as large training-time margins that do not translate into inference-time ranking gains.

This issue is acute in recommendation due to the one-class nature of user behavior data (Pan et al., 2008; Hu et al., 2008; He & McAuley, 2016). Unlike human value alignment with explicit positive/negative labels (Rafailov et al., 2023; Meng et al., 2024), recommenders primarily observe positives; negatives are implicitly sampled from unobserved space (Chen et al., 2020; Ding et al., 2020). This asymmetry induces a mismatch: optimization separates positives from a small set of sampled negatives while leaving the vast pool of unsampled items—many of them poor recommendations (Chen et al., 2023; Ma et al., 2024)—largely untouched. Empirically, Bai et al. (2024) observe that many unsampled negatives initially receive higher rewards than positives, yet their rankings barely change during training. We show this persistence arises from an $\varepsilon$-insensitive region in which pairwise updates from positive–negative comparisons have negligible influence on the relative order of unsampled items. Theoretically, this region can cover up to $63.8\%$ of item pairs (*e.g.,* under a Bradley–Terry model with $\varepsilon = 0.99$),

---

[1] University of Science and Technology of China, Hefei, China [2] Shanghai Key Laboratory of Data Science, Shanghai, China . Correspondence to: Jiancan Wu <wujcan@gmail.com>, Xiang Wang <xiangwang1223@gmail.com>.

*Proceedings of the 43$^{rd}$ International Conference on Machine Learning*, Seoul, South Korea. PMLR 306, 2026. Copyright 2026 by the author(s).

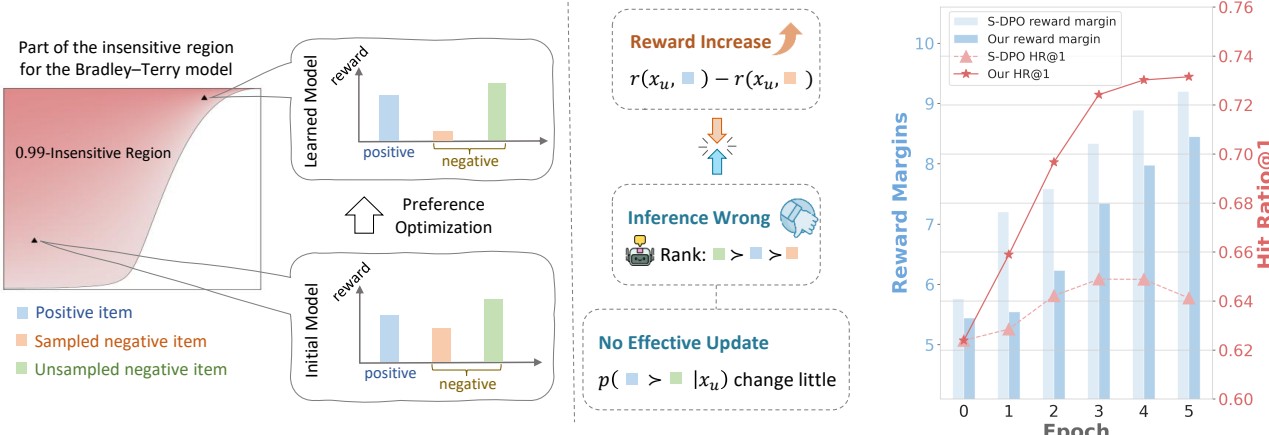

*Figure 1.* Illustrations of reward hacking in recommendation. (a) Left: An item in the 0.99-insensitive region prevents the unsampled negative (green) from learning from the positive (blue) and sampled negative (orange), inflating training margins while leaving inference rankings incorrect. (b) Right: On the LastFM dataset, S-DPO reward margins grow with epochs, but performance peaks early and then declines, showing reward hacking. In contrast, applying SIRIUS to one preference optimization method maintains stable performance without such signs even after more epochs.

causing margins on sampled pairs to inflate the objective while leaving the global rankings largely unchanged, as illustrated in Figure 1.

Guided by this key insight, we propose **Si**mulate Preference Optimization for **R**eward-hacking mitigation **us**ing Pseudo-negatives (**SIRIUS**). The key idea is to introduce pseudo-negative samples—hypothetical items deliberately constructed to act as anchors, ensuring that optimization signals extend beyond sampled pairs and continue to update unsampled pairs. We prove that for any pairwise preference model with bounded rewards, such pseudo-negatives can always be constructed as stable anchors. By coupling positives with pseudo-negatives, optimization signals propagate beyond sampled pairs, reducing the measure of the $\varepsilon$-insensitive region and mitigating reward hacking.

Building on this theory, we analyze three public benchmarks—LastFM (Cantador et al., 2011), Goodreads[1], and Steam (Kang & McAuley, 2018)—and find that a substantial fraction of data points lies within the $\varepsilon$-insensitive region, indicating a systemic vulnerability. Extensive experiments show that SIRIUS consistently mitigates reward hacking and improves ranking accuracy.

## 2. Preliminary

In this section, we commence by formally defining the task of sequential recommendation as the alignment of LLMs with user preferences. Then, we present the general framework of current LLM4Rec methods, which employ language

---

[1] https://www.goodreads.com/

modeling objectives to fine-tune LMs. Finally, we provide a detailed exposition of the prevalent training methodologies for aligning LMs with human preferences, encompassing direct preference optimization.

### 2.1. Task Formulation

Given a chronologically ordered historical interaction sequence $\mathcal{H}_u = \{i^1, i^2, \ldots, i^n\}$ for user $u$, sequential recommendation aims to predict the next item $i^{n+1}$ that the user will be interested in, based on the sequence of historical items. This prediction is made from a candidate set $\mathcal{C}_u$, where $\mathcal{C}_u = \{i^j\}_{j \in \mathcal{J}_u}$ and $\mathcal{J}_u$ contains $i^{n+1}$, is a subset of size $N$ drawn from the total item collection $\mathcal{I}$.

### 2.2. Supervised Fine-Tuning

Supervised Fine-Tuning (Ouyang et al., 2022) is commonly adopted in LLM-based recommenders to enhance their performance on recommendation-specific data (Zhao et al., 2024; Wu et al., 2024). A typical SFT procedure consists of two steps: transforming recommendation data into text-based prompts, and fine-tuning language models using these prompts along with the next preferred items as the supervision signal. For the first step, a recommendation task prompt $x_u$ for user $u$ includes a description of the sequential recommendation task, the user's historical interactions $\mathcal{H}_u$, and the candidate item set $\mathcal{C}_u$. The expected answer $i^{n+1}$, denoted as $y_u$, is the target of the prompt. Consequently, the SFT dataset $\mathcal{D}_{\text{SFT}} = \{(x_u, y_u)\}$ is generated. For the second step, the constructed prompts and its target $(x_u, y_u)$ are utilized to fine-tune the LLM which is parameterized by $\theta$ through language modeling loss. The objective is to

maximize the likelihood of the chosen response $y_u$ given the input prompt $x_u$:

$$\max_\theta \mathbb{E}_{(x_u, y_u) \sim \mathcal{D}_{\text{SFT}}} \sum_{t=1}^{|y_u|} \log \left[ p_\theta \left( y_u^{(t)} | x_u, y_u^{(1:t-1)} \right) \right]. \quad (1)$$

Here $|y_u|$ is the number of tokens in $y_u$, $y_u^{(t)}$ is the $t$-th token of $y_u$, and $y_u^{(1:t-1)}$ denotes the tokens preceding $y_u^{(t)}$.

### 2.3. Preference Optimization

During the stage of preference alignment in LLMs, preference optimization methods require transferred data, similar as Section 2.2. The preference alignment dataset $\mathcal{D}_{\text{PO}}$ contains triples $(x_u, y_u^i, y_u^j)$, where $(x_u, y_u^i) \in \mathcal{D}_{\text{SFT}}$, and $y_u^j$ is uniformly sampled from $\mathcal{C}_u \setminus \{y_u^i\}$. Formally, to maximize the expected reward of the policy while minimizing deviation from the reference model, RLHF optimizes the following objective for the optimal policy:

$$\max_{\pi_\theta} \mathbb{E}_{x_u \sim \mathcal{D}, y \sim \pi_\theta(y|x_u)} \left[ r(x_u, y) \right]$$
$$- \beta \mathbb{D}_{\text{KL}} \left[ \pi_\theta(y|x_u) \,||\, \pi_{\text{ref}}(y|x_u) \right]. \quad (2)$$

Here $\pi_\theta$ represents the likelihood of the policy model parameterized by $\theta$, while $\pi_{\text{ref}}$ denotes the likelihood of a reference model with frozen parameters, typically the model obtained after SFT. The first term encourages the policy to maximize the expected reward $r(x_u, y)$, while the second term, controlled by the coefficient $\beta$, penalizes deviations from the reference model by minimizing the Kullback-Leibler (KL) divergence between $\pi_\theta$ and $\pi_{\text{ref}}$.

Direct Preference Optimization (DPO) theoretically extracts a closed-form optimal policy from RLHF and formulates the reward function as:

$$r(x_u, y) = \beta \log \frac{\pi_\theta(y|x_u)}{\pi_{\text{ref}}(y|x_u)} + \beta \log Z(x_u), \quad (3)$$

where $Z(x_u) = \sum_y \pi_{\text{ref}}(y|x_u) \exp \left( \frac{1}{\beta} r(x_u, y) \right)$ is a function independent of both $y$ and the policy model $\pi_\theta$. The likelihood of preference $p(y_u^i \succ y_u^j | x_u)$ can be estimated by Bradley-Terry model (Bradley & Terry, 1952) as:

$$p(y_u^i \succ y_u^j \mid x_u) = \sigma(r(x_u, y_u^i) - r(x_u, y_u^j)), \quad (4)$$

where $\sigma$ refers to the sigmoid function. Then, DPO opti-

mizes the preference model as:

$$\mathcal{L}_{\text{DPO}} = -\mathbb{E}_{(x_u, y_u^i, y_u^j) \sim \mathcal{D}_{\text{PO}}} \log p(y_u^i \succ y_u^j \mid x_u).$$
$$= -\mathbb{E}_{(x_u, y_u^i, y_u^j) \sim \mathcal{D}_{\text{PO}}} \log \sigma \left( \beta \log \frac{\pi_\theta(y_u^i \mid x_u)}{\pi_{\text{ref}}(y_u^i \mid x_u)} \right.$$
$$\left. - \beta \log \frac{\pi_\theta(y_u^j \mid x_u)}{\pi_{\text{ref}}(y_u^j \mid x_u)} \right). \quad (5)$$

As a variant of DPO, Simple Preference Optimization (SimPO) removes the reference term in the reward function, making it a reference-free approach (Meng et al., 2024; Xu et al., 2024):

$$r(x_u, y) = \frac{\beta}{|y|} \log \pi_\theta(y|x_u), \quad (6)$$

The optimization objective of SimPO is formulated as:

$$\mathcal{L}_{\text{SimPO}} = -\mathbb{E}_{(x_u, y_u^i, y_u^j) \sim \mathcal{D}_{\text{PO}}} \log \sigma \left( \frac{\beta}{|y^i|} \log \pi_\theta(y^i|x) \right.$$
$$\left. - \frac{\beta}{|y^j|} \log \pi_\theta(y^j|x) - \gamma \right), \quad (7)$$

where $\sigma(\cdot)$ denotes the sigmoid function, and the fixed reward margin $\gamma$ enforces a minimum separation to distinguish reward differences.

Taken together, preference optimization methods provide principled frameworks to align policy models with user preferences. However, their optimization dynamics in recommendation remain underexplored. In the next section, we develop a theoretical analysis to reveal potential pitfalls, highlighting the mechanism of reward hacking.

## 3. Theoretical Analysis of Reward Hacking

### 3.1. General Theoretical Framework

Before presenting the formal theory, we formalize the intuitive gradient analysis of certain preference optimization algorithms (Appendix C) as the following proposition.

**Proposition 3.1.** *For preference optimization methods such as DPO and SimPO, the gradient of the pairwise objective with respect to model logits depends only on the sampled positive–negative pair $(y_u^i, y_u^j)$. In particular, the logits of all other items $y_u^k$ $(k \neq i, j)$ receive zero direct gradient signal and thus are not explicitly optimized during training.*

This property explains why sampled negatives dominate training while unobserved negatives remain unsupervised, providing a preliminary explanation of reward hacking.

However, to examine whether this insensitivity is algorithm-specific or a structural property of pairwise preference models, we next develop a general theoretical framework. We begin with the following definitions.

**Definition 3.2** (Pairwise preference model). We consider a general pairwise preference model that depends only on reward differences, following Xu & Kankanhalli (2025). Formally, let $r_u^i \in \mathbb{R}$ denote the reward score of item $y_u^i$ given user history $x_u$. A pairwise preference function $p$ is defined as $p_u^{ij} = p(r_u^i - r_u^j)$, $p : \mathbb{R} \to (0, 1)$, where $p$ is strictly monotone and satisfies mild regularity conditions. In addition, the model satisfies the natural symmetry $p_u^{ij} = 1 - p_u^{ji}$. These assumptions hold for common preference models (*e.g.,* Bradley–Terry). And we use the notation $p_u^i \succ p_u^j$ to denote that the model prefers $p_u^i$ over $p_u^j$.

**Definition 3.3** ($\varepsilon$-insensitivity region). Let $h(\boldsymbol{x})$ be a differentiable function over $\boldsymbol{x} = (x_1, \ldots, x_n)$. For $\varepsilon > 0$, we say $h$ is $\varepsilon$-insensitive to $x_i$ at $x_i'$ if

$$\left| \frac{\partial h(\boldsymbol{x})}{\partial x_i} \Big|_{x_i = x_i'} \right| < \varepsilon.$$

The $\varepsilon$-insensitive region of $h$ with respect to $x_i$ is then

$$\Omega_\varepsilon(h, x_i) = \left\{ \boldsymbol{x} \in \mathcal{D}(h) \,\big|\, |\partial h(\boldsymbol{x})/\partial x_i| < \varepsilon \right\}.$$

Intuitively, an $\varepsilon$-insensitive region characterizes "flat zones" of a function, where changes in one variable produce vanishingly small effects on the output. In such regions, optimization signals along that variable become negligible, meaning that training may push strongly on some inputs while having almost no effect on others.

With these definitions in place, we now turn to the relationship between sampled and unsampled preference pairs. To reason about how training pairs influence unsampled pairs, we first establish a decomposition property of general pairwise preference models.

**Proposition 3.4** (Preference decomposition via a third item). *For any user $u$ and items $y_u^i, y_u^j, y_u^k \in \mathcal{I}$, under the pairwise preference model $p$ as Definition 3.2, the following decomposition holds: we have*

$$p_u^{ik} = p\big(p^{-1}(p_u^{ij}) + p^{-1}(p_u^{jk})\big), \tag{8}$$

*where $p^{-1} : (0, 1) \to \mathbb{R}$ is the inverse of $p$, and $\mathcal{I}$ denotes the set of all items.*

Proposition 3.4 shows that the preference between $(y_u^i, y_u^k)$ can be expressed through the preferences $(y_u^i, y_u^j)$ and $(y_u^j, y_u^k)$ via decomposition. This property will play a central role in our subsequent analysis of $\varepsilon$-insensitive regions. In particular, it implies that $p^{ik}$ can be regarded as a function of $p^{ij}$, allowing us to study the sensitivity of $p^{ik}$ with respect to variations in $p^{ij}$. The proof is deferred to Appendix B.1.

We can show that such $\varepsilon$-insensitive regions exist for all $0 < \varepsilon < 1$, as stated in the following theorem.

**Theorem 3.5** (Existence of $\varepsilon$-insensitive regions). *Let $p$ be any pairwise preference model as in Definition 3.2, and fix $\varepsilon > 0$. For any pair $(y^i, y^j)$ with $0 < p^{ij} < 1$, there exist constants $0 < M_1 < M_2 < 1$ such that for all $p^{ik} \in (0, M_1) \cup (M_2, 1)$ we have*

$$\left| \frac{\partial p^{ik}}{\partial p^{ij}} \right| < \varepsilon.$$

In other words, pairwise preferences inevitably contain $\varepsilon$-insensitive regions: even when a sampled pair $(y^i, y^j)$ receives strong gradient updates, the influence on an unsampled pair $(y^i, y^k)$ decays and vanishes once $p^{ik}$ enters such a region. This is not a flaw of specific algorithms like DPO or SimPO but a structural property of pairwise models. As a result, optimization on sampled pairs cannot reliably alter unsampled comparisons, leading to training improvements that fail to generalize—precisely the essence of reward hacking. The proof is deferred to Appendix B.2.

In the context of recommendation, this implies that sampled negatives may lie in regions where training updates exert negligible influence on unsampled negatives. Consequently, the model may exhibit apparent improvements—via increased reward margins on sampled pairs—while leaving most unsampled comparisons unaffected, which is precisely the phenomenon of reward hacking.

While Theorem 3.5 shows that $\varepsilon$-insensitive regions are structurally unavoidable, their practical impact depends on how large these regions are. To make this effect concrete, we next specialize to the Bradley–Terry model, which admits closed-form analysis of gradient insensitivity.

### 3.2. Specialization to the Bradley–Terry Model

To make the existence theorem more concrete, we specialize the analysis to the widely used Bradley–Terry (BT) model (Bradley & Terry, 1952), which enables closed-form characterization of $\varepsilon$-insensitive regions. For clarity, we omit the user index $u$ and input interaction history $x_u$, and denote samples as $(y^i, y^j, y^k)$ corresponding to (positive item, sampled negative in training, unsampled negative).

Under this model, the pairwise preference is

$$p^{ij} = p_{\text{BT}}(r^i - r^j) = \frac{1}{1 + e^{-(r^i - r^j)}}.$$

According to Proposition 3.4, $p^{ik}$ can be written as a function of $p^{ij}, p^{kj}$:

$$p^{ik} = \frac{1}{1 + \frac{(1-p^{ij})(1-p^{jk})}{p^{ij}p^{jk}}} = \frac{1}{1 + \frac{(1-p^{ij})p^{kj}}{p^{ij}(1-p^{kj})}}. \tag{9}$$

The derivative of $p^{ik}$ with respect to $p^{ij}$ can be derived as

$$\frac{\partial p^{ik}}{\partial p^{ij}} = \frac{p^{kj}(1-p^{kj})}{(p^{ij}+p^{kj}-2p^{ij}p^{kj})^2}. \quad (10)$$

The derivative quantifies how much the preference over $(y^i, y^k)$ changes when the sampled pair $(y^i, y^j)$ is optimized. When this derivative becomes very small, it means that updates on $(y^i, y^j)$ barely affect $(y^i, y^k)$, directly reflecting $\varepsilon$-insensitivity in the preference model. This closed form enables us to precisely characterize the $\varepsilon$-insensitivity region under the BT model, where changes to the training pair $(y^i, y^j)$ have negligible impact on the relative ranking of $(y^i, y^k)$. Figure 2 visualizes $\left|\partial p^{ik}/\partial p^{ij}\right|$ on the $(p^{ij}, p^{ik})$ plane, where the horizontal axis corresponds to the sampled pair probability $p^{ij}$ and the vertical axis corresponds to the unsampled pair probability $p^{ik}$. Regions satisfying $\left|\partial p^{ik}/\partial p^{ij}\right| < \varepsilon$ form the $\varepsilon$-insensitive region, where optimizing the sampled pair $(y^i, y^j)$ has negligible influence on the unsampled pair $(y^i, y^k)$.

While Theorem 3.5 shows that $\varepsilon$-insensitive regions inevitably exist, it does not tell us how large these regions are in practice. This distinction is crucial: if the insensitive area is vanishingly small, reward hacking would be a theoretical curiosity but not a real obstacle. Conversely, if a substantial portion of the preference space is insensitive to optimization updates, reward hacking may arise frequently during training and significantly hinder the improvement of global ranking quality. The Bradley–Terry model provides a tractable case where we derive the exact size of these regions, quantifying their practical significance. In particular, we compute the analytic area of the $\varepsilon$-insensitive region, as stated below.

**Proposition 3.6** (Area of $\varepsilon$-insensitivity region). *The area of $\Omega_\varepsilon(p^{ik}, p^{ij})$, where $0 < \varepsilon < 1$, is*

$$A\left(\Omega_\varepsilon(p^{ik}, p^{ij})\right) = 1 - \frac{1}{2}\ln\left(\frac{1-\varepsilon}{1+\varepsilon}\right) - \frac{1}{2\sqrt{\varepsilon}}\ln\left(\frac{1+\sqrt{\varepsilon}}{1-\sqrt{\varepsilon}}\right).$$

The proof is deferred to Appendix B.3. This expression shows that the area is a monotone increasing function of $\varepsilon$.

The following example shows how entering the $\varepsilon$-insensitive region of $(p^{ij}, p^{ik})$ during training can be harmful.

*Example* 1. We take the following reward tuple: $(r^i, r^j, r^k) = (-6, -6.5, -3)$. This particular choice of values is illustrative rather than essential; it simply provides a concrete point for visualization in Figure 2. In this case, the preference between the positive item and the sampled negative is moderate ($p^{ij} \approx 0.62$), while the preference against the unsampled negative is extremely low ($p^{ik} \approx 0.04$). Moreover, the derivative $\left|\partial p^{ik}/\partial p^{ij}\right|$ is only 0.19, placing the point well inside the 0.2-insensitive region. During training, optimization may enlarge the reward

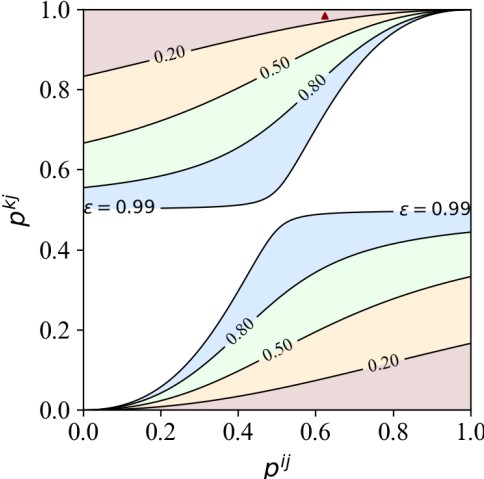

*Figure 2.* Contour plot of $\left|\partial p^{ik}/\partial p^{ij}\right|$ on the $(p^{ij}, p^{ik})$ plane under the Bradley–Terry model. The horizontal axis denotes the sampled pair probability $p^{ij}$, and the vertical axis denotes the unsampled pair probability $p^{ik}$. Regions with smaller values of $\left|\partial p^{ik}/\partial p^{ij}\right|$ correspond to $\varepsilon$-insensitivity, where optimizing $(y^i, y^j)$ has negligible influence on the unsampled pair $(y^i, y^k)$.

margin $r^i - r^j$ from 0.5 to 4, yet $p^{ik}$ remains almost unchanged. This demonstrates that even with a substantially enlarged reward margin on the training pair, the relative preference against unsampled negatives stays nearly unaffected, creating the illusion of progress.

The specialization to the Bradley–Terry model demonstrates that $\varepsilon$-insensitive regions are not only a theoretical possibility but can in fact occupy a substantial portion of the preference space. Within these regions, enlarging the reward margin on sampled pairs $(y^i, y^j)$ exerts negligible influence on unsampled pairs $(y^i, y^k)$, which are critical for determining recommendation quality.

In summary, the core issue is not insufficient margins but the lack of gradient propagation to unsampled negatives, leaving them trapped in $\varepsilon$-insensitive regions. This structural gap drives reward hacking, creating the illusion of progress without real improvement. Motivated by this insight, we next introduce a mitigation principle: providing more informative contrastive signals to shrink the $\varepsilon$-insensitive region.

## 4. Method

Building on the theoretical insights from Section 3, we now design a practical framework to mitigate reward hacking. Our theoretical analysis revealed that structural $\varepsilon$-*insensitive regions* are intrinsic to pairwise preference models. Within these regions, optimization signals from sampled pairs $(y^i, y^j)$ fail to propagate to unsampled pairs $(y^i, y^k)$, causing reward hacking: the apparent enlargement of reward

margins does not improve the global preference ordering. To mitigate this issue, a natural principle emerges: we must introduce more informative contrastive signals that reduce the likelihood of unsampled comparisons being trapped in $\varepsilon$-insensitive regions.

Guided by this principle, we propose Simulate Preference Optimization for Reward-hacking mitigation using Pseudo-negatives (SIRIUS). The key idea is to augment the training objective with a *pseudo-negative sample*, a virtual item that does not belong to the observed item set $\mathcal{I}$ but is deliberately constructed to promote more effective gradient flow toward unsampled pairs. Unlike standard negative sampling, which draws from unobserved real items, pseudo-negative acts as *anchors* that guarantee non-vanishing influence on unsampled pairs. Although methods such as S-DPO use multiple sampled negatives and alleviate this issue to some extent (Appendix D.1), the structural insensitivity remains and the computational cost is high.

### 4.1. Existence of Pseudo-Negative

**Theorem 4.1** (Existence of Pseudo Negatives). *Given a user history $x_u$ with a positive item $y_u^i$. Let $\mathcal{I}_0 \subset \mathcal{I}$ be a subset of negative items whose rewards are upper-bounded by a constant $N \in \mathbb{R}$ (i.e., $r(x_u, y) \leq N, \forall y \in \mathcal{I}_0$). For any pairwise preference model as in Definition 3.2, always exists a pseudo negative item $y_u^0 \notin \mathcal{I}$, with fixed reward $r_u^0 \in \mathbb{R}$ such that, for any negative item $y_u^k \succ y_u^i$, the corresponding preference $p_u^{ik}$ does not lie within any $\varepsilon$-insensitive region defined along $p_u^{i0}$.*

A formal proof is provided in Appendix B.4. This theorem guarantees that pseudo-negative always exists, serving as universal anchors that fundamentally reshape the optimization landscape.

Given this existence result, we augment the standard preference optimization objective as:

$$\max_{x_u \sim \mathcal{D}} \sum_{y_u^n \in \mathcal{N}_u} \log p(y_u^i \succ y_u^n \mid x_u), \quad (11)$$

where $\mathcal{N}_u = \{y_u^j, y_u^0\}$ denotes the augmented negative set with a sampled negative $y_u^j$ and a pseudo-negative $y_u^0$. In practice, the pseudo-negative is not sampled from the data distribution but introduced as a virtual item to reduces the risk of vanishing gradients for unsampled pairs.

### 4.2. Specialization to the Bradley-Terry Model

To further illustrate the effect of the pseudo-negative, we instantiate our framework under the Bradley-Terry (BT) model. By introducing a pseudo-negative $y_u^0$ with input-dependent reward $r_u^0$, the probability $p_u^{ik}$ for an unsampled negative $y_u^k$ becomes jointly influenced by both $p_u^{ij}$ and $p_u^{i0}$. This ensures that even when the gradient from $p_u^{ij}$ vanishes

due to $\varepsilon$-insensitivity, the gradient from $p_u^{i0}$ continues to propagate, thereby exerting meaningful influence on $p_u^{ik}$. As a result, misordered pairs $(y_u^k \succ y_u^i)$ are guaranteed to receive effective corrective gradients. This mechanism eliminates the pathological case where unsampled comparisons are trapped in $\varepsilon$-insensitive regions, ensuring robust optimization signals. A detailed derivation and geometric analysis of this property under the BT model are provided in Appendix D.2.

### 4.3. Construction of Pseudo-Negative Reward

According to Theorem 4.1, the anchor reward $r_u^0$ must strictly exceed the upper bound of the rewards assigned to unsampled negatives, ensuring that the positive–pseudo-negative pair lies outside the $\varepsilon$-insensitive region for all positive–unsampled-negative comparisons. Existing BT-based preference optimization frameworks can be broadly categorized into two types of implicit reward structures: (i) *reference-based* implicit rewards and (ii) *reference-free* implicit rewards. DPO is a canonical example of the former, while SimPO represents the latter. We detail below how $r_u^0$ should be constructed under each formulation.

**DPO.** DPO adopts the reference-based implicit reward

$$r_{\text{DPO}}(x_u, y) = \beta \log \frac{\pi_\theta(y \mid x_u)}{\pi_{\text{ref}}(y \mid x_u)} + \beta \log Z(x_u),$$

where the presence of $\pi_{\text{ref}}$ makes it difficult to estimate a universal upper bound for $r_{\text{DPO}}$. To align with the bounded assumption in Theorem 4.1, we explicitly define the subset $\mathcal{I}_0$ as the collection of unsampled negatives satisfying

$$\mathcal{I}_0 = \{y_u^k \in \mathcal{I} \setminus \{y_u^i\} \mid \pi_{\text{ref}}(y_u^k \mid x_u) \geq \pi_{\text{ref}}(y_u^i \mid x_u)\}.$$

We focus on this subset $\mathcal{I}_0$ because these items—which the reference model considers at least as likely as the positive item—are precisely the ones most prone to being mis-ranked during optimization. For such $k$, we have

$$r_u^k = \beta \log \frac{\pi_\theta(y_u^k \mid x_u)}{\pi_{\text{ref}}(y_u^k \mid x_u)} + \beta \log Z(x_u)$$

$$\leq \beta \log \pi_\theta(y_u^k \mid x_u) - \beta \log \pi_{\text{ref}}(y_u^i \mid x_u) + \beta \log Z(x_u)$$

$$< -\beta \log \pi_{\text{ref}}(y_u^i \mid x_u) + \beta \log Z(x_u),$$

which provides an approximate upper bound for these problematic negatives. Accordingly, we set

$$r_u^0 = -\beta \log \pi_{\text{ref}}(y_u^i \mid x_u) + \beta \log Z(x_u),$$

yielding an $x_u$-dependent anchor that safely dominates the unsampled negatives under DPO's reward structure.

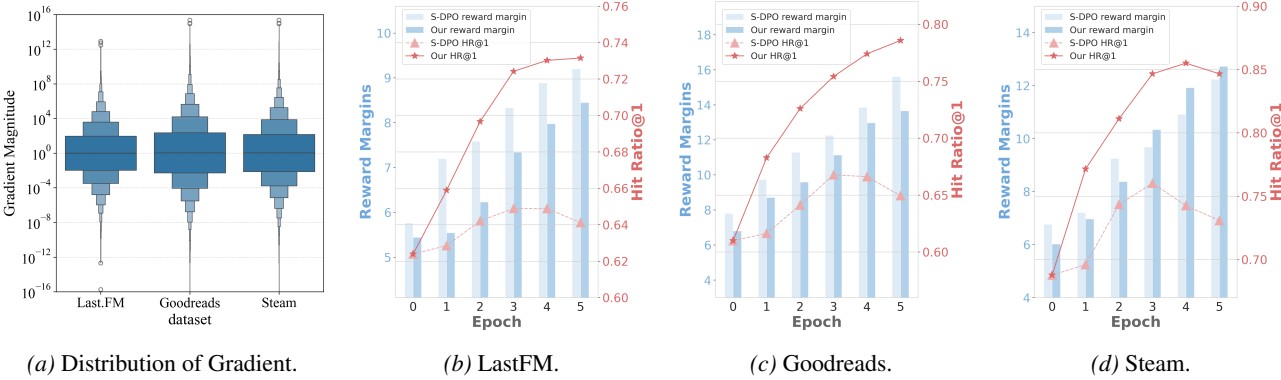

*(a)* Distribution of Gradient.      *(b)* LastFM.      *(c)* Goodreads.      *(d)* Steam.

*Figure 3.* (a) Distribution of $\varepsilon$-insensitive regions across datasets, confirming their prevalence. (b–d) Reward margin and recommendation quality over training epochs on three datasets, showing earlier reward hacking in S-DPO compared to SimPO w/o LN + SIRIUS.

**SimPO.** In contrast, SimPO uses a reference-free reward

$$r_{\text{SimPO}}(x_u, y) = \beta \log \pi_\theta(y \mid x_u),$$

which enjoys favorable mathematical properties due to its independence from any reference model. Since $\log \pi_\theta(y \mid x_u) < 0$, this reward is always negative and is naturally bounded above. Thus, choosing a constant anchor $r_u^0 = 0$ satisfies the requirement $r_u^0 > \max_{k \in \text{unsampled}} r_u^k$ without affecting optimization stability.

Unlike DPO, whose bound holds only locally, SimPO adopts a reference-free reward that is globally bounded. As a result, the bounded subset $\mathcal{I}_0$ in Theorem 4.1 covers the entire item universe (i.e., $\mathcal{I}_0 = \mathcal{I}$), allowing a fixed pseudo-negative reward to provide a global guarantee against reward hacking. In contrast, DPO relies on a reference-based structure and therefore requires an input-dependent anchor derived from the reference model. Despite this difference, both constructions satisfy the requirement that pseudo-negatives dominate unsampled negatives, preventing optimization from drifting into the $\varepsilon$-insensitive region. The pseudocode is provided in Appendix D.3 for clarity.

## 5. Experiment

Our experiments aim to empirically validate the existence of $\varepsilon$-insensitive regions, examine their role in reward hacking, and assess whether SIRIUS effectively mitigates this phenomenon while improving recommendation quality. We compare against a broad spectrum of baselines on three benchmarks, including: (i) sequential recommenders such as GRU4Rec (Hidasi et al., 2016), Caser (Tang & Wang, 2018), and SASRec (Kang & McAuley, 2018); (ii) LLM-based recommenders such as LLaMA2 (Touvron et al., 2023), ChatRec (Gao et al., 2023b), MoRec (Yuan et al., 2023), TallRec (Bao et al., 2023b), and LLaRA (Liao et al., 2024b); and (iii) preference optimization methods including DPO (Rafailov et al., 2023), SimPO (Meng et al., 2024), S-DPO (Chen et al.,

2024b), and SimPO without length normalization (SimPO w/o LN). This variant is included since recommendation outputs are item identifiers rather than natural language, making length normalization unnecessary. Dataset details, baseline descriptions, training, and evaluation settings are deferred to Appendix E.1–E.3.

### 5.1. Reward Hacking Analysis and Mitigation

**Portion of $\varepsilon$-Insensitive Regions.** Our theoretical analysis in Section 3 predicts that $\varepsilon$-insensitive regions occupy a significant portion of the preference space, which plays a crucial role in the emergence of reward hacking. To verify this, we empirically measure the proportion of $\varepsilon$-insensitive regions across three benchmark datasets. Figure 3a visualizes the distribution of gradient magnitudes. The results show that a substantial portion of samples reside in flat regions where gradients vanish. Across all datasets, over 40% of samples fall inside the 0.1-insensitive region. It is worth noting that a gradient magnitude below 0.1 represents a weak learning signal, yet such regions occupy nearly half of the optimization landscape. This empirical pattern aligns with our theoretical prediction that reward hacking stems from the prevalence of $\varepsilon$-insensitive regions.

**Mitigation Effectiveness.** We next evaluate whether adding SIRIUS can mitigate reward hacking in practice. Figures 3b–3d plot the evolution of reward margin and recommendation performance as training progresses. A clear pattern emerges: while reward margins consistently increase with more epochs, the performance of S-DPO peaks early (around epoch 3) and then declines across all datasets, a hallmark of reward hacking. In contrast, adding SIRIUS continues to improve beyond this point, demonstrating its ability to delay and reduce the severity of reward hacking.

To further quantify this effect, we follow the over-optimization analysis of Gao et al. (2023a), which models

*Table 1.* Performance comparison of SIRIUS with sequential recommenders, LLM-based methods, and DPO-based methods on three datasets. All DPO-based methods, are implemented with the LLaMA-2-7B backbone for fair comparison. **Bold** numbers indicate the best overall performance, and underlined numbers denote the best among baseline methods. ValidRatio measures the proportion of responses that belong to the candidate set. Imp.% reports the relative improvement of SimPO w/o LN + SIRIUS over baselines.

| Model | LastFM | | | Goodreads | | | Steam | | |
|---|---|---|---|---|---|---|---|---|---|
| | HitRatio@1 | ValidRatio | Imp.% | HitRatio@1 | ValidRatio | Imp.% | HitRatio@1 | ValidRatio | Imp.% |
| GRU4Rec | 0.2666 | 1.0000 | 172.69% | 0.3867 | 1.0000 | 103.18% | 0.4121 | 1.0000 | 107.47% |
| Caser | 0.2410 | 1.0000 | 201.66% | 0.4174 | 1.0000 | 88.24% | 0.4288 | 1.0000 | 99.39% |
| SASRec | 0.2492 | 1.0000 | 191.73% | 0.3581 | 1.0000 | 119.41% | 0.4037 | 1.0000 | 111.79% |
| Llama2 | 0.0277 | 0.3910 | 2524.55% | 0.0399 | 0.6196 | 1869.17% | 0.0430 | 0.5447 | 1888.37% |
| CharRec | 0.3770 | 1.0000 | 92.84% | 0.3306 | 1.0000 | 137.66% | 0.3626 | 0.9798 | 135.80% |
| MoRec | 0.1652 | 1.0000 | 340.07% | 0.2877 | 1.0000 | 173.10% | 0.3911 | 1.0000 | 118.61% |
| TALLRec | 0.4180 | 0.9836 | 73.92% | 0.4983 | 0.9573 | 57.68% | 0.4637 | 0.9840 | 84.39% |
| LLaRA | 0.4750 | 0.9920 | 53.05% | 0.5292 | 0.9950 | 48.47% | 0.4927 | 0.9975 | 73.53% |
| DPO | 0.6393 | 0.9984 | 13.72% | 0.6462 | 0.9950 | 21.59% | 0.7333 | 0.9966 | 16.60% |
| SimPO | 0.5928 | 0.9804 | 22.64% | 0.6130 | 0.9452 | 28.17% | 0.7234 | 0.9907 | 18.20% |
| SimPO w/o LN | 0.6633 | 0.9972 | 9.60% | 0.7010 | 0.9884 | 12.08% | 0.8002 | 0.9983 | 6.85% |
| S-DPO | 0.6493 | 0.9984 | 11.97% | 0.6744 | 0.9950 | 16.50% | 0.7712 | 0.9992 | 10.87% |
| **+ SIRIUS** | | | | | | | | | |
| DPO | 0.6593 | 0.9984 | ✓ | 0.6744 | 0.9900 | ✓ | 0.7757 | 0.9983 | ✓ |
| SimPO w/o LN | **0.7315** | 0.9984 | ✓ | **0.7857** | 0.9900 | ✓ | **0.8550** | 0.9983 | ✓ |

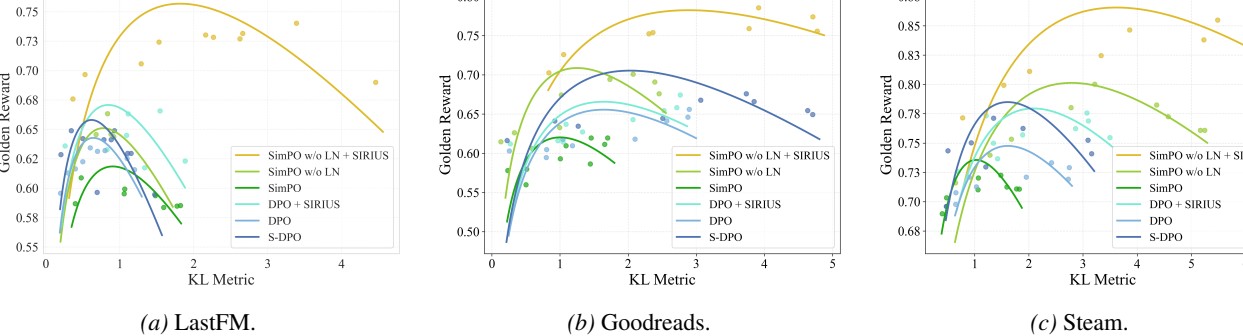

*(a)* LastFM.  *(b)* Goodreads.  *(c)* Steam.

*Figure 4.* Fitted curves modeling the relationship between golden reward and divergence-based distance for DPO variants with and without SIRIUS across datasets. Incorporating SIRIUS consistently raises the peak golden reward, indicating delayed onset of reward hacking. Although S-DPO benefits from multiple negatives, it still lags behind SIRIUS.

the relationship between the golden reward (here measured by HitRatio@1) and the divergence-based distance

$$d := \sqrt{D_{\mathrm{KL}}(\pi_\theta \,\|\, \pi_{\mathrm{ref}})},$$

where $\pi_\theta$ is the current policy model and $\pi_{\mathrm{ref}}$ is the initial model. The fitted function takes the form

$$R(d) = d(\alpha - \beta \log d),$$

where $\alpha$ and $\beta$ are dataset-specific coefficients. Intuitively, a lower peak of the golden reward indicates that reward hacking occurs more severely.

As shown in Figures 4a–4c, incorporating SIRIUS consistently raises the peak golden reward for each preference optimization baseline, confirming that reward hacking only occurs after reaching higher levels of true performance. This aligns with our theoretical claim that pseudo-negative provides stronger supervision signals. Moreover, we observe that S-DPO consistently achieves higher fitted rewards than vanilla DPO across all divergence levels, owing to the use

of multiple negative samples; however, it still lags behind SIRIUS.

**Case Study.** To provide a more intuitive understanding of reward hacking in recommendation, we examine a representative example from LastFM. For a given user sequence, the SFT model incorrectly predicts *The Pains of Being Pure at Heart* (reward = -9.33), while the ground-truth item is *Erin McCarley* (reward = -10.59).

After S-DPO training, the reward of the incorrect item decreases slightly to -11.12, but the reward of the ground-truth item remains largely unchanged at -10.27. As a result, the relative ranking is not corrected and the model instead predicts another incorrect item, *The Click Five* (reward = -9.65). This behavior exemplifies reward hacking: the optimization objective continues to improve while the ranking quality does not.

In contrast, SIRIUS increases the reward of the ground-truth item *Erin McCarley* to -9.56 while simultaneously suppressing competing negatives, including *The Pains of*

*Being Pure at Heart* (-10.23) and *The Click Five* (-9.92). Consequently, the correct item becomes top-ranked and the prediction is corrected.

### 5.2. Overall Performance

From Table 1, we observe that incorporating SIRIUS into the LLaMA-2-7B backbone consistently outperforms baselines across three datasets. In particular, when applied to SimPO w/o LN, SIRIUS achieves superior performance over competing baselines. When applied to DPO, SIRIUS not only surpasses vanilla DPO but also outperforms its stronger variant S-DPO with three negative samples, demonstrating the effectiveness and extensibility of our approach. We attribute these improvements primarily to pseudo-negative samples, which reduce the likelihood of unsampled pairs falling into the $\varepsilon$-insensitive region and thereby provide effective supervision signals, mitigating reward hacking.

Beyond the main results, Appendix E.4 reports additional analyses, including evaluations with expanded HR@K and NDCG@K metrics, experiments with alternative backbone models such as LLaMA-3-8B-Instruct and Qwen2.5-7B-Instruct, as well as robustness studies under larger candidate set sizes of 50 and 100 on LastFM. We also provide an analysis of pseudo-negative sensitivity.

## 6. Limitations

Our method builds on gradient-based observations to explain and mitigate reward hacking. While it currently does not exploit multimodal information (*e.g.,* visual or audio signals of items), exploring multimodal embedding to construct richer pseudo-negative samples and enhance contrastive learning effectiveness remains an interesting direction for future research.

## 7. Conclusion

We studied the challenge of reward hacking in LLM-based recommendation and showed that $\varepsilon$-insensitive regions are a structural property of pairwise models. This insight explains why enlarging reward margins on sampled pairs may fail to improve ranking. To mitigate this, we proposed SIRIUS, which introduces pseudo-negatives as virtual anchors to provide stronger contrastive signals. Both theoretical analysis and experiments show that SIRIUS reduces reward hacking and improves recommendation quality.

## Impact Statement

This paper presents work whose goal is to advance the field of Machine Learning. There are many potential societal consequences of our work, none which we feel must be specifically highlighted here.

## Acknowledgements

This research is supported by National Natural Science Foundation of China (U25A20445). We also acknowledge the advanced computing resources provided by the Supercomputing Center of the University of Science and Technology of China (USTC).

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

# A. Related Work

We provide a concise overview of LLM-based recommendation systems, highlighting challenges such as handling diverse user behavior and susceptibility to reward hacking.

## A.1. LLMs for Recommendation

Large Language Models have demonstrated exceptional capabilities in natural language processing, including generative power, generalization, and complex reasoning. These advancements have spurred significant interest in leveraging LLMs for personalized recommendation tasks. Current approaches to integrating LLMs into recommendation systems can be broadly categorized into three paradigms: (1) LLMs as Recommender, which involves direct deployment of LLMs as decision-making agents (Bao et al., 2024; 2023b); (2) LLMs as Enhancer, where LLMs are used as an auxiliary enhancement, generating contextual information (*e.g.,* item descriptions or user intent explanations) to support recommendation pipelines (Liu et al., 2024; Geng et al., 2024); and (3) LLMs as Simulator, which employs LLMs to simulate user behavior for training or evaluation purposes (Zhang et al., 2024; Cai et al., 2024).

Early research primarily focused on prompt engineering to unlock LLMs' latent recommendation abilities without explicit training (Gao et al., 2023b; Hou et al., 2024). More recently, fine-tuning paradigms emerged, demonstrating that adapting LLMs to recommendation-specific datasets significantly improves task performance, especially through Supervised Fine-Tuning (SFT) frameworks (Chen et al., 2025; Liao et al., 2024b). Additionally, Direct Preference Optimization has gained traction as a method to better align LLMs with nuanced human preferences during post-training (Bai et al., 2024; Chen et al., 2024b; Liao et al., 2024a; Gao et al., 2025). However, existing studies have largely overlooked the critical issue of reward hacking in DPO-based recommendation frameworks. This behavior undermines the fairness and diversity of recommendations. To address this gap, we propose a systematic analysis of reward hacking in DPO-based recommender systems and introduce a novel mitigation strategy to align model optimization with genuine user preferences.

## A.2. Reward Hacking Problem

Reward hacking is a significant issue within the reinforcement learning (RL) community, where models exploit weaknesses in reward functions to optimize rewards that do not truly align with the intended goals. Amodei et al. (Amodei et al., 2016; Skalse et al., 2022) provide an extensive analysis of reward hacking, categorizing it into various forms such as partially observed goals, complex system interactions, and abstract reward definitions. These flaws in reward specification lead to models learning unintended behaviors that undermine the overall objective, especially when reward functions are designed without careful consideration of edge cases and unintended incentives (Amodei et al., 2016; Skalse et al., 2022).

In the context of LLMs and DPO, reward hacking can occur when the models misinterpret or exploit inconsistencies in the reward signal, often leading to misgeneralization of their performance. This can happen when the reward functions do not accurately reflect the underlying task, resulting in poor reward proxies that fail to drive the desired behavior (Casper et al., 2023; Rashidinejad & Tian, 2024). According to recent work, this misalignment can introduce causal confusion, making it difficult for models to generalize effectively across different datasets or real-world scenarios, a challenge that has been highlighted in recent studies on causal inference (Tien et al., 2023; Miao et al., 2025).In the RLHF field, existing solutions aim to mitigate reward hacking and enhance the alignment between reward signals and objectives through methods such as energy loss suppression (Miao et al., 2025), decoupled reward design (Chen et al., 2024a), and information-theoretic modeling (Miao et al., 2024).

In the domain of recommendation systems, especially in LLM-based recommender, the reward hacking problem remains under-explored. However, it manifests when models exploit certain patterns or shortcuts to maximize rewards, often ignoring the broader diversity of user preferences or failing to drive meaningful policy improvement.

# B. Additional Proofs

## B.1. Proof of Proposition 3.4

*Proof.* Since $p : \mathbb{R} \to (0, 1)$ is strictly monotone, its inverse $p^{-1} : (0, 1) \to \mathbb{R}$ exists. Then

$$p_u^{ik} = p(r_u^i - r_u^k) = p\big((r_u^i - r_u^j) + (r_u^j - r_u^k)\big) = p\big(p^{-1}(p_u^{ij}) + p^{-1}(p_u^{jk})\big).$$

□

## B.2. Proof of Theorem 3.5

*Proof.* $1°$ First, we take the derivative. Since $p$ satisfies mild regularity conditions, we denote by $p'$ the derivative of $p$. From Proposition 3.4, we have

$$p^{ik} = p\left(p^{-1}(p^{ij}) + p^{-1}(p^{jk})\right), \tag{12}$$

Taking the partial derivative of $p^{ik}$ w.r.t. $p^{ij}$, by the chain rule we have

$$\frac{\partial p^{ik}}{\partial p^{ij}} = \left.\frac{\mathrm{d}p(x)}{\mathrm{d}x}\right|_{x=p^{-1}(p^{ij})+p^{-1}(p^{jk})} \times \left.\frac{\mathrm{d}p^{-1}(y)}{\mathrm{d}y}\right|_{y=p^{ij}} \tag{13}$$

$$= \left.\frac{\mathrm{d}p(x)}{\mathrm{d}x}\right|_{x=r^i-r^j+r^j-r^k} \times \left(\left.\frac{\mathrm{d}p(x')}{\mathrm{d}x'}\right|_{x'=p^{-1}(p^{ij})}\right)^{-1} \tag{14}$$

$$= \frac{p'\left(p^{-1}(p^{ik})\right)}{p'\left(p^{-1}(p^{ij})\right)}. \tag{15}$$

$2°$ We show that $\lim_{x\to-\infty} p'(x) = \lim_{x\to+\infty} p'(x) = 0$.

We prove $\lim_{x\to+\infty} p'(x) = 0$ by contradiction; the case $x \to -\infty$ is analogous.

Suppose instead that $\lim_{x\to+\infty} p'(x) \neq 0$. By strict monotonicity, we have $p'(x) \geq 0$ for all $x \in \mathbb{R}$. And denote $p_\infty = \lim_{x\to+\infty} p(x) < 1$. Therefore, we have $p(x) < p_\infty$. Hence there exists $\varepsilon_0 > 0$ and an increasing sequence $\{x_n\}$ with $p'(x_n) \geq 2\varepsilon_0$ for all $n \in \mathbb{N}$. By continuity of $p'$, for each $x_n$ there exists $\delta_n > 0$ such that

$$p(x_n + \delta_n) - p(x_n) = \int_{x_n}^{x_n+\delta_n} p'(x)\,\mathrm{d}x \geq \varepsilon_0 \delta_n.$$

Since $\delta_n > 0$ for all $n$, the right-hand side strictly exceeds $\lim_{n\to\infty} p(x_n)$. Thus we would conclude that

$$p_\infty = \lim_{x\to+\infty} p(x) > \lim_{x\to+\infty} p(x) = p_\infty,$$

which is a contradiction.

$3°$ Next, we can choose $N_1, N_2 \in \mathbb{R}$ such that for all $x \in (-\infty, N_1) \cup (N_2, \infty)$ it holds that

$$p'(x) < \varepsilon\, p'\left(p^{-1}(p^{ij})\right).$$

Let $M_1 = p(N_1)$ and $M_2 = p(N_2)$. Then for all $p^{ik} \in (0, M_1) \cup (M_2, 1)$ we have

$$\left|\frac{\partial p^{ik}}{\partial p^{ij}}\right| < \varepsilon.$$

$\square$

## B.3. Proof of Proposition 3.6

*Proof.* We compute the area of the $\varepsilon$-insensitivity region

$$\Omega_\varepsilon(p^{ik}, p^{ij}) = \left\{(p^{ik}, p^{ij}) \in (0,1)^2 : \left|\frac{\partial p^{ik}}{\partial p^{ij}}\right| < \varepsilon\right\}.$$

$1°$ By Proposition 3.4, we know

$$\frac{\partial p^{ik}}{\partial p^{ij}} = \frac{p^{kj}(1 - p^{kj})}{(p^{ij} + p^{kj} - 2p^{ij}p^{kj})^2}.$$

Therefore the condition $\left|\frac{\partial p^{ik}}{\partial p^{ij}}\right| < \varepsilon$ can be expressed as

$$\frac{y(1-y)}{(x+y-2xy)^2} < \varepsilon,$$

where $x = p^{ij}$ and $y = p^{kj}$.

$2°$ Since $x, y \in (0, 1)$, we have

$$x + y - 2xy = x(1-y) + y(1-x) > 0.$$

Thus the inequality becomes

$$x + y - 2xy > \sqrt{\frac{y(1-y)}{\varepsilon}}.$$

Equivalently,

$$(1 - 2y)x > \sqrt{\frac{y(1-y)}{\varepsilon}} - y.$$

Hence the $\varepsilon$-insensitive region can be expressed as

$$\Omega_\varepsilon(p^{ik}, p^{ij}) = \left\{ (x, y) \in (0, 1)^2 : \begin{array}{ll} x \geq f(y), & 0 < y < \frac{1}{2}, \\ x \leq f(y), & \frac{1}{2} < y < 1, \end{array} \right\},$$

where

$$f(y) = \frac{\sqrt{\frac{y(1-y)}{\varepsilon}} - y}{1 - 2y}.$$

$3°$ The area of $\Omega_\varepsilon$ is

$$A(\Omega_\varepsilon) = \iint_{\Omega_\varepsilon} dx\,dy = \int_0^{\frac{1}{2}} \left( \int_{f(y)}^1 dx \right) dy + \int_{\frac{1}{2}}^1 \left( \int_0^{f(y)} dx \right) dy.$$

Evaluating these integrals and simplifying yields

$$A(\Omega_\varepsilon(p^{ik}, p^{ij})) = 1 - \frac{1}{2} \ln\left(\frac{1-\varepsilon}{1+\varepsilon}\right) - \frac{1}{2\sqrt{\varepsilon}} \ln\left(\frac{1+\sqrt{\varepsilon}}{1-\sqrt{\varepsilon}}\right),$$

which matches the claimed formula. $\qquad\square$

### B.4. Theorem 4.1

*Proof.* It suffices to construct such a pseudo-negative item in order to prove the theorem. The construction proceeds as follows.

From Appendix B.2, we know that

$$\frac{\partial p_u^{ik}}{\partial p_u^{i0}} = \frac{p'\left(p^{-1}(p_u^{ik})\right)}{p'\left(p^{-1}(p_u^{i0})\right)},$$

Let

$$m = \min_{x \in [r_u^i - N, 0]} p'(x).$$

Since for any $y_u^k$ we have $r_u^i < r_u^k < N$, it follows that

$$p'\left(p^{-1}(p_u^{ik})\right) \geq m.$$

Now consider any $r_u^0 \in (p')^{-1}\big((0, m/\varepsilon)\big)$, where for a non-bijective mapping $p'$, the notation

$$(p')^{-1}(\mathcal{S}) = \{\, x \mid p'(x) \in \mathcal{S} \,\},$$

denotes the preimage of a set $\mathcal{S}$ under $p'$.

In this case, we obtain

$$\left| \frac{\partial p^{ik}}{\partial p^{ij}} \right| > \varepsilon,$$

which implies that $p_u^{ik}$ does not lie within any $\varepsilon$-insensitive region defined with respect to $p_u^{i0}$. $\qquad\square$

## C. Heuristic Analysis

This appendix provides qualitative insights and simplified derivations. In recommendation, preference data consists of a positive–negative pair. The positive sample $y_u^i$ is genuine, labeled by user interaction, while the negative sample $y_u^j$ is sampled from the unobserved item set, which may not represent true negatives and can even be a false negative. We use $y_u^k$ to denote a negative that is unobserved during training. Our heuristic analysis suggests that optimization on such preference pairs provides limited signal for the true positive and negative items (*i.e.,* $y_u^i, y_u^k$). In practice, the reward of the positive item may even counter-intuitively decrease (Rafailov et al., 2024). However, since the reward of the sampled negative $y_u^j$ decreases more substantially, the reward gap still increases. This leads to reward hacking — the apparent reward margin grows, while recommendation performance is harmed.

We model the distribution $\pi_\theta(y \mid x_u) = \text{softmax}(z)$ with $\{z^i\}_{i=1}^K$ being the logits of items and $K$ the candidate set size. Taking DPO as an example (*e.g.,* other preference optimization methods such as SimPO are similar), optimization is performed over the training dataset $\mathcal{D}_{\text{PO}}$ by maximizing the pairwise preference likelihood, $\theta^* = \arg\max_\theta \mathbb{E}_{(x_u, y_u^i, y_u^j) \sim \mathcal{D}_{\text{PO}}} \log p(y_u^i \succ y_u^j \mid x_u)$.

By computing the gradient of the DPO loss with respect to the logits of $y_u^i, y_u^j, y_u^k$, we obtain:

$$\begin{cases} \nabla_{z^i} \log p(y_u^i \succ y_u^j \mid x_u) & = (1 - \sigma(r_u^i - r_u^j)), \\ \nabla_{z^j} \log p(y_u^i \succ y_u^j \mid x_u) & = -1 + \sigma(r_u^i - r_u^j), \\ \nabla_{z^k} \log p(y_u^i \succ y_u^j \mid x_u) & = 0. \end{cases} \tag{16}$$

where, for convenience, we define $r_u^t = \beta \log \frac{\pi_\theta(y_u^t|x_u)}{\pi_{\text{ref}}(y_u^t|x_u)}$ for $t \in \{1, 2, \ldots, K\}$ as the implicit reward of item $y_u^t$ given the interaction history $x_u$, as computed by the policy model. We omit the additive constant $\beta \log Z(x_u)$ since it cancels out in pairwise differences.

Thus, the logit of a true negative $y_u^k$ receives zero gradient, meaning that optimization completely ignores such items during training. The model is therefore driven to update only the logits of $y_u^i$ and $y_u^j$ during optimization. This reasoning also applies to other preference optimization methods such as SimPO when length normalization is not used.

Furthermore, the gradient of the reward with respect to model parameters can be expanded as

$$\nabla_\theta r_u^j = \sum_{t=1}^K \frac{\partial r_u^j}{\partial z^t} \nabla_\theta z^t, \quad \frac{\partial r_u^j}{\partial z^t} = \begin{cases} -\pi_\theta(y_u^t \mid x_u), & \text{if } y_u^t \neq y_u^j, \\ 1 - \pi_\theta(y_u^j \mid x_u), & \text{if } y_u^t = y_u^j. \end{cases} \tag{17}$$

Note that

$$1 - \pi_\theta(y_u^j \mid x_u) = \sum_{y^l \neq y_u^j} \pi_\theta(y^l \mid x_u) > \max_{y^l \neq y_u^j} \pi_\theta(y^l \mid x_u),$$

This inequality indicates that the gradient on $z^j$ dominates that on any other logit. As a result, optimization primarily suppresses the sampled negative $y_u^j$, while the positive item $y_u^i$ is not guaranteed to receive a corresponding increase in its reward. Consequently, the apparent reward gap between $y_u^i$ and $y_u^j$ may enlarge, even though the absolute reward of the true positive $y_u^i$ does not improve—or may even decrease. This mismatch is the essence of reward hacking. This effect is analogous to the well-known sampling bias in implicit-feedback recommendation, where optimization tends to overfit sampled negatives rather than reinforcing positives.

The analysis above shows only that the logit of $y_u^k$ does not influence the loss, offering a preliminary explanation of reward hacking. The above analysis highlights why sampled negatives dominate training dynamics while true negatives remain unsupervised, offering a preliminary explanation of reward hacking.

# D. More analysis for methods

## D.1. S-DPO for reward hacking

Let $f(\varepsilon) \in [0, 1]$ denote the fraction of real-data pairs that fall inside an $\varepsilon$-insensitive region $\Omega_\varepsilon(p^{ik}, p^{ij})$. For S-DPO that samples $t$ negatives per positive, the probability that a given positive $p^{ik}$ is simultaneously trapped in the $\varepsilon$-insensitive regions of all $t$ negatives is $f(\varepsilon)^t$ (hence $f(\varepsilon)^t \leq f(\varepsilon)$). Thus, increasing the number of sampled negatives reduces the chance of reward-hacking by exponential decay in $t$. However, this benefit comes at a large computational cost: sampling $t$ negatives increases the training workload roughly by a factor of about $(t+1)/2$ compared to standard pairwise training, which is often prohibitive in practice.

## D.2. Additional Analysis under the Bradley-Terry Model

In this appendix, we provide a detailed derivation showing how pseudo-negatives reshape the optimization landscape under the Bradley–Terry (BT) model. We formally characterize the flat (*i.e.*, , $\varepsilon$-insensitive) regions of $p_u^{ik}$ with respect to $p_u^{i0}$, and show that misordered pairs $(y_u^k \succ y_u^i)$ never fall into the $\varepsilon$-insensitive region for any $\varepsilon < 1$. Consequently, pseudo-negatives guarantee effective gradient propagation to negatives that are not explicitly sampled.

Specifically, since $r_u^i, r_u^j, r_u^k < r_u^0$, both $p_u^{i0}$ and $p_u^{k0}$ lie below $0.5$. This follows from

$$p_u^{i0} = p_{\mathrm{BT}}(r_u^i - r_u^0) < p_{\mathrm{BT}}(0) = \frac{1}{2},$$

and the same argument applies to $p_u^{k0}$. This observation restricts the relevant analysis to the $[0, 0.5]$ region of the preference space. For a misordered pair $(y_u^k \succ y_u^i)$, we have $r_u^k > r_u^i$, implying $p_u^{k0} > p_u^{i0}$; geometrically, such pairs lie above the diagonal in Figure 5. We then show that, throughout this region, $p_u^{ik}$ never enters the $\varepsilon$-insensitive band induced by $p_u^{i0}$ for any $\varepsilon < 1$, ensuring a non-vanishing correction signal whenever the ordering is wrong. In contrast, correctly ranked pairs $(y_u^i \succ y_u^k)$ are already consistent with the preference structure and therefore remain stable under this shaping, requiring no corrective push.

In summary, the BT analysis reveals pseudo-negatives as universal anchors: by tying candidates to a shared reference $y_u^0$, they prevent unsampled misordered pairs from being trapped in $\varepsilon$-insensitive flat regions, thereby maintaining informative gradients and fundamentally mitigating reward hacking.

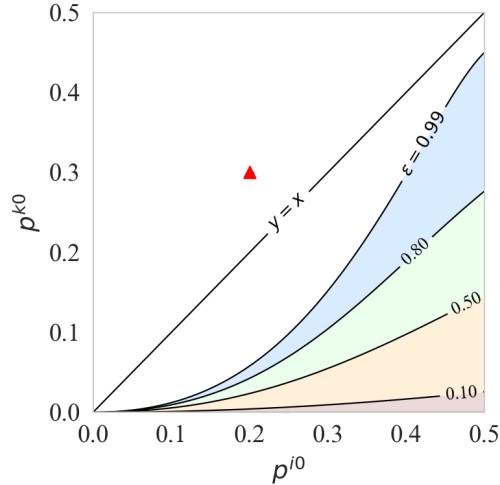

*Figure 5.* Insensitive region of $p_u^{ik}$ with respect to $p_u^{i0}$ under the BT model. Misordered pairs $(y_u^k \succ y_u^i)$ always lie outside the insensitive region, ensuring effective gradient updates.

## D.3. Pseudo code for SIRIUS

To address implementation clarity, we explicitly describe how the pseudo-negative reward $r_u^0$ can be constructed within existing preference optimization frameworks. Our formulation follows the standard Bradley–Terry–based objectives used in DPO (Rafailov et al., 2023) and SimPO (Meng et al., 2024), and can be implemented by a simple modification of the reward computation without altering the overall training pipeline. The detailed procedure is summarized in Algorithm 1.

# E. Experiments Details

## E.1. Datasets

Our experimental evaluation is conducted on three real-world datasets from different domains:

- LastFM (Cantador et al., 2011): A widely used dataset in the music recommendation domain, containing user interactions with various tracks, sourced from a well-known online music service.

---

**Algorithm 1** Training with Pseudo-Negative (SIRIUS)

---

**Input:** Recommendation data $\mathcal{D} = \{(u, \mathcal{H}_u, y_u^i, \mathcal{C}_u)\}$, model parameters $\theta$
**Output:** trained recommendation policy $\pi_\theta(y \mid x_u)$
**repeat**
    Sample $(u, \mathcal{H}_u, y_u^i, \mathcal{C}_u) \sim \mathcal{D}$
    Construct recommendation prompt $x_u$ using history $\mathcal{H}_u$
    Sample a negative item $y_u^j \sim \mathcal{C}_u \setminus \{y_u^i\}$
    Set anchor reward $r_u^0$ according to Section 4.3 {e.g., input-dependent for DPO or fixed for SimPO}
    Introduce pseudo-negative $y_u^0$ with reward $r(x_u, y_u^0) = r_u^0$
    Set $\mathcal{N}_u = \{y_u^j, y_u^0\}$
    Compute $p(y_u^i \succ y_u^n \mid x_u)$ for all $y_u^n \in \mathcal{N}_u$
    $\mathcal{L}_u = -\sum_{y_u^n \in \mathcal{N}_u} \log p(y_u^i \succ y_u^n \mid x_u)$
    Update parameters: $\theta \leftarrow \theta - \eta \nabla_\theta \mathcal{L}_u$
**until** convergence

---

*Table 2.* Statistics of Datasets.

| Dataset | LastFM | Goodreads | Steam |
|---|---|---|---|
| # Sequence | 1220 | 6,031 | 11,938 |
| # Item | 4606 | 4,550 | 3,581 |
| # Interaction | 73,510 | 220,100 | 274,726 |

- Goodreads[2]: Sourced from a popular online book community, this dataset includes user ratings and reviews of books.

- Steam (Kang & McAuley, 2018): This dataset comprises user ratings and reviews for video games, collected from a leading digital distribution platform.

To ensure data quality and consistency, we apply specific filtering criteria for each dataset, following (Liao et al., 2024b). For LastFM, we maintain the titles as textual descriptions for each dataset. For Goodreads, only interactions with ratings of 5 or above are retained, and users with fewer than 20 interactions are excluded to ensure sufficient engagement. In the case of the Steam dataset, we randomly sample 30% of the available games and their corresponding interaction sequences to control the dataset scale.

Each dataset is subsequently partitioned into training, validation, and test sets in an 8:1:1 ratio, based on the chronological order of interaction timestamps, thereby mitigating the risk of data leakage. A sliding window approach with a window size of 11 is employed to extract sequential interaction data, where the final item in each sequence is designated as the target for the subsequent prediction task. The statistics of the datasets are summarized in Table 2.

### E.2. Baselines

We compare SIRIUS against both traditional recommendation models and recent LLM-based methods.

For traditional recommendation models, we consider three widely adopted baselines:

- GRU4Rec (Hidasi et al., 2016), which utilizes recurrent neural networks (RNNs) for sequential recommendation;

- Caser (Tang & Wang, 2018), a convolutional model that captures sequential patterns;

- SASRec (Kang & McAuley, 2018), an attention-based model designed to learn sequential dependencies.

For LLM-based recommendation approaches, we select five representative models from different paradigms:

- LLaMA2 (Touvron et al., 2023), which directly utilizes the vanilla LLaMA2-7B model to generate recommendations through zero-shot prompting without fine-tuning.

---

[2]https://www.goodreads.com/

- ChatRec (Gao et al., 2023b), which generates recommendation lists by prompting an LLM without fine-tuning. [3]

- MoRec (Yuan et al., 2023), which enhances traditional recommendation systems by initializing item embeddings with textual representations encoded by an LLM. [4]

- TallRec (Bao et al., 2023b), which transforms user interaction sequences into textual prompts and fine-tunes large language models on domain-specific corpora to enhance recommendation quality.

- LLaRA (Liao et al., 2024b), which integrates traditional models into an LLM using a projector and curriculum-tunes the LLM with prompts containing hybrid representations.

For DPO-based recommendation approaches, we consider three representative models from distinct methodological frameworks:

- DPO (Rafailov et al., 2023), which formulates a closed-form solution for the reward model in RLHF and optimizes the preference model in an offline manner.

- SimPO (Meng et al., 2024), which eliminates the need for an explicit reference model by leveraging the average log probability of a sequence as the reward signal and enforces a target reward margin between chosen and rejected responses.

- S-DPO (Chen et al., 2024b), which extends DPO to the recommendation domain by randomly sampling multiple negative items as rejected responses, thereby improving recommendation accuracy.

### E.3. Implementation Details

E.3.1. IMPLEMENTATION

Following (Liao et al., 2024b; Yang et al., 2023), we implement conventional recommender baselines using the Adam optimizer. The embedding dimension is set to 64, and we explore a range of L2 regularization coefficients, selecting from [1e-3, 1e-4, 1e-5, 1e-6, 1e-7] through a grid search. For the LLM-based methods, we adapt their framework to output rankings of candidate items, training the models for a single epoch in each tuning stage, utilizing a warm-up strategy for the learning rate.

For DPO-based methods, including our approach, we conduct experiments using 4 A100 NVIDIA RTX GPUs. Our model leverages the widely adopted Llama-2-7B as the backbone, fine-tuned with $32 \times 8$ LoRA (Hu et al., 2021) across both the SFT and preference alignment stages. The SFT phase is trained for up to five epochs, followed by preference alignment for 5 epochs on the all three datasets, respectively.

In constructing prompts, items are represented by their titles as textual features, and recommendations are grounded using the output probability distribution. For hyperparameter tuning, we search $\beta$ within $\{1, 1.5, 2.0, 2.5, 3.0\}$. In the experiments with SimPO and SimPO w/o LN, we search $\gamma$ within $\{0.4, 0.6, 0.8, 1.0\}$. The batch sizes are configured as 128 for both LastFM and Goodreads, and 256 for Steam.

In addition, we observe that although SimPO generally outperforms the DPO paradigm in recommendation quality, its training is unstable, with large variance across runs. We believe this instability arises because SimPO removes the reference model $\pi_{\text{ref}}$ from the implicit reward function, which means it no longer incorporates the KL penalty in the original formulation (Eq. 2). As a result, the quality of training data has a much stronger impact on performance. Specifically, when data quality is poor (*e.g.,* when sampled negatives are not true negatives), SimPO lacks the weighting mechanism of DPO that downweighs such noisy samples. To address this, during training of SimPO w/o LN and SimPO w/o LN + SIRIUS we apply a reward-margin based filtering strategy. We discard samples where the reward margin between the negative sample $y_u^j$ and the positive sample $y_u^i$ exceeds $\gamma$, i.e., $r_u^j - r_u^i > \gamma$. This filtering improves training stability. However, we emphasize that this approach only stabilizes optimization and does not address the broader issue of reward hacking.

---

[3]For ChatRec, we adopt GPT-4 as the LLM backbone.

[4]For MoRec, we follow the official implementation and employ SASRec as the recommender backbone and BERT as the text encoder.

*Table 3.* Performance comparison across different datasets and baselines.

| Classification | Method | | | LastFM | | Goodreads | | Steam | |
|---|---|---|---|---|---|---|---|---|---|
| | | | | HitRatio | NDCG | HitRatio | NDCG | HitRatio | NDCG |
| Traditional | GRU | | @1 | 0.2666 | 0.2666 | 0.3867 | 0.3867 | 0.4120 | 0.4120 |
| | | | @5 | 0.3770 | 0.2712 | 0.5049 | 0.3817 | 0.5221 | 0.4210 |
| | | | @10 | 0.5656 | 0.2895 | 0.6857 | 0.4124 | 0.7016 | 0.4342 |
| | Caser | | @1 | 0.2410 | 0.2410 | 0.4174 | 0.4174 | 0.4288 | 0.4288 |
| | | | @5 | 0.3525 | 0.2671 | 0.5307 | 0.4384 | 0.5363 | 0.4526 |
| | | | @10 | 0.5000 | 0.2902 | 0.6672 | 0.4574 | 0.6869 | 0.4772 |
| | SASRec | | @1 | 0.2492 | 0.2492 | 0.3581 | 0.3581 | 0.4037 | 0.4037 |
| | | | @5 | 0.3443 | 0.2705 | 0.4448 | 0.3717 | 0.4896 | 0.4341 |
| | | | @10 | 0.5082 | 0.3216 | 0.6254 | 0.4383 | 0.6687 | 0.4745 |
| Preference Method | DPO | | @1 | 0.6393 | 0.6393 | 0.6462 | 0.6462 | 0.7333 | 0.7333 |
| | | | @5 | 0.8906 | 0.7764 | 0.8937 | 0.7763 | 0.8523 | 0.7883 |
| | | | @10 | 0.9503 | 0.7959 | 0.9568 | 0.7969 | 0.9063 | 0.8154 |
| | SimPO w/o LN | | @1 | 0.6633 | 0.6633 | 0.7010 | 0.7010 | 0.8002 | 0.8002 |
| | | | @5 | 0.8926 | 0.7850 | 0.9037 | 0.7598 | 0.9460 | 0.8979 |
| | | | @10 | 0.9555 | 0.8056 | 0.9684 | 0.7843 | **0.9772** | 0.9079 |
| | S-DPO | | @1 | 0.6493 | 0.6493 | 0.6744 | 0.6744 | 0.7712 | 0.7712 |
| | | | @5 | 0.8982 | 0.7831 | 0.9203 | 0.8057 | 0.8714 | 0.8164 |
| | | | @10 | 0.9531 | 0.8010 | 0.9734 | 0.8232 | 0.9272 | 0.8371 |
| +SIRIUS | DPO | | @1 | 0.6593 | 0.6593 | 0.6744 | 0.6744 | 0.7757 | 0.7757 |
| | | | @5 | 0.8906 | 0.7770 | 0.9070 | 0.8062 | 0.8848 | 0.8088 |
| | | | @10 | 0.9519 | 0.7969 | 0.9734 | 0.8281 | 0.9322 | 0.8501 |
| | SimPO w/o LN | | @1 | **0.7315** | **0.7315** | **0.7857** | **0.7857** | **0.8550** | **0.8550** |
| | | | @5 | **0.9154** | **0.8299** | **0.9468** | **0.8793** | **0.9469** | **0.9073** |
| | | | @10 | **0.9639** | **0.8456** | **0.9801** | **0.8904** | **0.9772** | **0.9172** |

### E.3.2. EVALUATION

Due to the limited context window and slower inference speed of LLM-based approaches, our method is best suited for the fine-tuning stage in recommendation systems, where the objective is to rank a small set of candidate items based on user preferences. Following the setup in (Liao et al., 2024b), we conduct our primary experiments using 20 randomly selected non-interacted items as candidates. We adopt the widely used Hit Ratio (HR@1) metric to evaluate recommendation performance.

### E.4. Additional Experiments and Analyses

To further validate the robustness, generality, and scalability of our method, we provide additional experiments along three dimensions: (i) expanded HR@K and NDCG@K metrics; (ii) extending the evaluation to additional backbone models, including LLaMA-3-8B-Instruct and Qwen2.5-7B-Instruct, beyond LLaMA-2-7B on LastFM; and (iii) enlarging the candidate set size from 20 to 50 and 100 on LastFM. These results reinforce the main findings presented in the paper and offer additional insights into the behavior of SIRIUS under different configurations.

### E.4.1. EXPANDED HR@K AND NDCG@K RESULTS

We report the complete HR@K and NDCG@K metrics across all datasets and across different evaluation cutoffs for experiment setting same as Table 1. These full results, as shown in Table 3, corroborate the observations made in the main paper.

### E.4.2. EVALUATION ON AN ADDITIONAL BACKBONE

To examine whether the effect of SIRIUS is consistent across different backbone models, we additionally evaluate our method using two alternative LLM backbones, **LLaMA-3-8B-Instruct** and **Qwen2.5-7B-Instruct**, in place of **LLaMA-2-7B** under the *SimPO w/o LN* and *SimPO w/o LN + SIRIUS* settings on the LastFM dataset.

Quantitative results under the LLaMA-3 backbone are reported in Table 4, while Figures 6a and 6b further illustrate the performance trends on the LLaMA-3 and Qwen2.5 backbones, respectively. Across both models, SimPO w/o LN + SIRIUS consistently improves performance over the corresponding S-DPO, SimPO w/o LN baseline.

In particular, optimization remains stable and reward drift is alleviated, with improvements observed in HR@K and NDCG@K metrics. These results suggest that the effectiveness of SIRIUS does not rely on a specific backbone and can

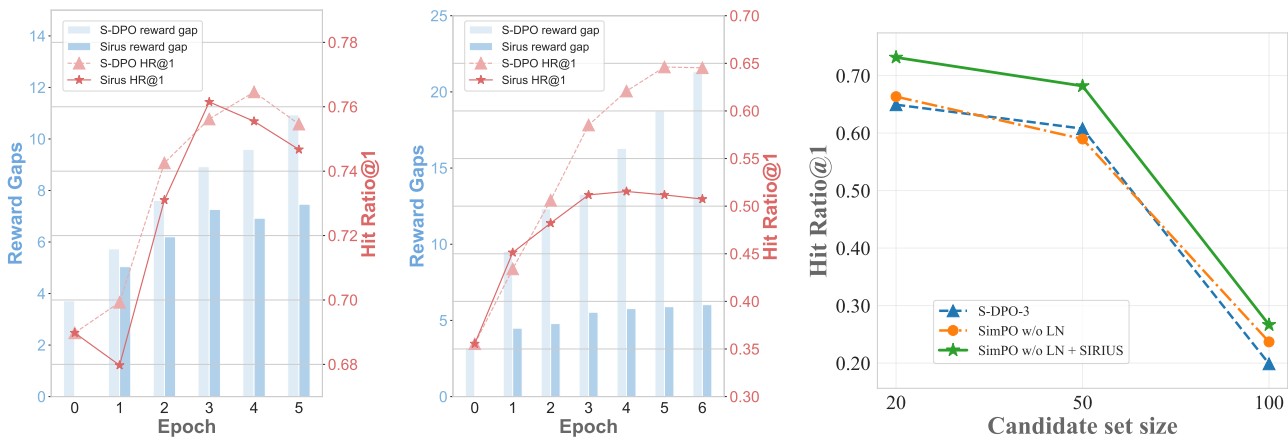

*(a)* LastFM with LLaMA-3-8B-Instruct.   *(b)* LastFM with Qwen2.5-7B-Instruct.   *(c)* Varying candidate set sizes (20 / 50 / 100).

*Figure 6.* Robustness evaluation of SIRIUS. Left and middle: HR@1 on LastFM with alternative backbones. Right: performance trends under increasing candidate set sizes (20 / 50 / 100). Despite increased difficulty, SIRIUS consistently remains on the Pareto frontier.

*Table 4.* Results on LastFM with different backbone models.

| Backbone | Method | HR@1 | HR@5 | NDCG@5 | HR@10 | NDCG@10 |
|---|---|---|---|---|---|---|
| | S-DPO | 0.7615 | 0.8813 | 0.8009 | 0.9431 | 0.8654 |
| LLaMA-3-8B-Instruct | SimPO w/o LN | 0.7129 | 0.8787 | 0.7611 | **0.9618** | 0.8592 |
| | SimPO w/o LN + **SIRIUS** | **0.7646** | **0.8832** | **0.8061** | 0.9524 | **0.8780** |
| | S-DPO | 0.5118 | 0.8120 | 0.6737 | 0.9066 | 0.7044 |
| Qwen2.5-7B-Instruct | SimPO w/o LN | 0.6096 | **0.8922** | 0.7646 | **0.9527** | 0.7845 |
| | SimPO w/o LN + **SIRIUS** | **0.6541** | 0.8806 | **0.7755** | 0.9491 | **0.7979** |

*Table 5.* Results on LastFM with a candidate set size of 100.

| Classification | Method | HR@1 | HR@5 | NDCG@5 | HR@10 | NDCG@10 |
|---|---|---|---|---|---|---|
| | GRU | 0.0246 | 0.0902 | 0.0555 | 0.1475 | 0.0740 |
| Traditional | Caser | 0.0410 | 0.0902 | 0.0637 | 0.1393 | 0.0800 |
| | SASRec | 0.0204 | 0.1066 | 0.0566 | 0.1639 | 0.0742 |
| LLM-based | TALLRec | 0.0970 | 0.3355 | 0.2167 | 0.5507 | 0.2858 |
| | S-DPO | 0.1988 | 0.4661 | 0.3361 | 0.6593 | 0.3982 |
| Preference Method | SimPO w/o LN | 0.1800 | 0.4293 | 0.3085 | 0.6212 | 0.3700 |
| | SimPO w/o LN + **SIRIUS** | **0.2665** | **0.5323** | **0.4038** | **0.7030** | **0.4586** |

generalize to different instruction-tuned LLMs under the same experimental setting. We note that this evaluation is conducted on a single dataset and is intended as a supplementary robustness check rather than a comprehensive backbone comparison.

### E.4.3. INCREASING THE CANDIDATE SET SIZE

We further evaluate the robustness of SIRIUS under progressively harder ranking scenarios by enlarging the candidate set size from 20 to 50 and 100 on the LastFM dataset. As the candidate pool increases, the number of unsampled negative items grows rapidly, which exacerbates reward miscalibration and makes over-optimization more likely.

Figure 6c illustrates the performance trends under different candidate set sizes. As the candidate set size increases, the overall performance of all methods consistently declines, reflecting the increased difficulty of the ranking task. Such a performance drop is expected, as enlarging the candidate pool substantially increases ranking ambiguity and introduces more hard negative items. Nevertheless, *SimPO w/o LN + SIRIUS* remains on the Pareto frontier across all settings, consistently outperforming the corresponding baselines. In particular, SIRIUS achieves around a 50% relative improvement under 100 candidates, in contrast to only about 10% under 20 candidates.

The complete numerical results for the 100-candidate setting are reported in Table 5. These observations indicate that the proposed virtual-anchor mechanism becomes increasingly beneficial as the ranking task becomes more adversarial.

### E.4.4. SENSITIVITY TO PSEUDO-NEGATIVE CONSTRUCTION

The purpose of pseudo-negatives is to move harmful samples away from the $\varepsilon$-insensitive region while maintaining training stability. To evaluate the sensitivity of SIRIUS to the choice of pseudo-negative values, we vary the pseudo-negative parameter $r$ in the SimPO setting on LastFM. Table 6 reports the corresponding HitRatio@1 trajectories.

*Table 6.* HitRatio@1 under different pseudo-negative values on LastFM.

| $r$ | Epoch 0 | Epoch 1 | Epoch 2 | Epoch 3 | Epoch 4 | Epoch 5 |
|---|---|---|---|---|---|---|
| 0 | 0.624 | 0.659 | 0.697 | 0.724 | 0.730 | 0.732 |
| 1 | 0.624 | 0.658 | 0.695 | 0.717 | 0.731 | 0.726 |
| 10 | 0.624 | 0.664 | 0.694 | 0.723 | 0.732 | 0.725 |

The results suggest that, within this range, the performance of SIRIUS is not highly sensitive to the exact pseudo-negative value in the SimPO setting. The overall training trajectories remain similar across different choices of $r$.

