# OpenReview forum: "Mitigating Reward Hacking in LLM-based Recommendation: A Preference Optimization Approach"
_ICML.cc/2026/Conference — ICML 2026 regular_

### Official Review · Reviewer_c2o4 · 2026-03-11

**Soundness:** 3
**Presentation:** 2
**Significance:** 3
**Originality:** 3
**Overall Recommendation:** 4
**Confidence:** 3

**Summary:**

This paper studies reward hacking in LLM-based recommendation under preference optimization. It introduces the ε-insensitive region to explain why improving sampled pairwise rewards may not improve global ranking, and proposes SIRIUS, which uses pseudo-negative anchors to improve optimization beyond observed pairs. Experiments on three benchmarks show consistent gains and suggest the method effectively mitigates this issue. Overall, the paper addresses an important problem with a clear idea.

**Compliance With Llm Reviewing Policy:**

Affirmed.

**Final Justification:**

Reward hacking in LLM-based recommendation is an interesting and timely topic. However, the rebuttal offers limited additional information and does not materially affect my evaluation, so I am keeping my original score.

**Key Questions For Authors:**

1. Could the authors clarify what is truly recommendation-specific in SIRIUS, versus what would transfer to general preference optimization? This would help me better assess the novelty of the paper.
2. Could the authors include one  concrete recommendation examples showing: (i) how reward hacking manifests for a user/candidate set under standard preference optimization, and (ii) how SIRIUS changes the ranking?

**Limitations:**

Yes

**Strengths And Weaknesses:**

Strengths

1. The paper studies reward hacking in recommendation, an important and timely problem. Framing this issue in LLM-based recommendation with pairwise preference optimization is meaningful.
2. A key strength is the novel theoretical framing via the ε-insensitive region, which helps explain why better sampled-pair rewards may not improve global ranking.
3. The paper has a strong theoretical foundation, with analysis for general pairwise preference models and the Bradley–Terry setting.

Weaknesses

1. The evaluation is limited, mainly focusing on ranking results and training dynamics. Broader metrics would strengthen the empirical support.
2. The proposed solution appears not fully recommendation-specific; its uniqueness to this domain is not yet fully convincing.
3. The paper is theory-heavy and would benefit from more concrete recommendation examples to improve intuition and practical interpretability.

---

> ### Author Rebuttal · Authors · 2026-03-31
>
> Thank you for your thoughtful review and constructive suggestions.
>
> > Q1: The evaluation is limited, mainly focusing on ranking results and training dynamics. Broader metrics would strengthen the empirical support.
> >
> > A1: We agree that broader empirical views can further strengthen the paper. At the same time, our current evaluation is centered on the three aspects most directly connected to our claims: final ranking performance, the reward-hacking dynamics during training, and the empirical proportion/distribution of samples that fall into the ε-insensitive region. In particular, the last quantity provides direct evidence that the phenomenon we analyze is not only theoretical but also prevalent in practice. We will make this empirical support clearer in the revision.
>
> > Q2: Could the authors clarify what is truly recommendation-specific in SIRIUS, versus what would transfer to general preference optimization?
> >
> > A2: We agree that this distinction should be made clearer. The core anchor-based idea in SIRIUS may be transferable to broader preference-optimization settings. What is recommendation-specific in our work is the structural problem we focus on: recommendation requires improving global ranking quality over a large candidate or item space, rather than only improving pairwise comparisons among a small number of candidates. In many RLHF-style settings, the final evaluation is often based on win rate over a small number of compared responses, whereas recommendation requires selecting and ranking the truly relevant item from a much larger candidate set. In this setting, the key challenge is whether optimizing an observed pair can effectively propagate to many unseen items in the ranking space. SIRIUS is designed specifically to address this weak-propagation problem in recommendation, which is also why our analysis and evaluation are centered on ranking behavior over large candidate sets.
>
> > Q3: The paper is theory-heavy and would benefit from more concrete recommendation examples to improve intuition and practical interpretability. Could the authors include one concrete recommendation example showing how reward hacking manifests and how SIRIUS changes the ranking?
> >
> > A3: We appreciate this suggestion and agree that a more concrete recommendation example improves intuition. Beyond the illustrative example already included in the current version, we provide the following case study on LastFM. For one sequence, the SFT model incorrectly predicts *The Pains of Being Pure at Heart* (reward = `-9.33`) instead of the correct item *Erin McCarley* (`-10.59`). After S-DPO training, the model slightly reduces the reward of the incorrect item to `-11.12`, but leaves the reward of the correct item largely unchanged at `-10.27`, so the relative ranking is still not corrected and the model instead predicts another wrong item, *The Click Five* (`-9.65`). This is exactly the type of reward hacking we discuss. In contrast, SIRIUS increases the reward of *Erin McCarley* to `-9.56` while demoting negatives such as *The Pains of Being Pure at Heart* (`-10.23`) and *The Click Five* (`-9.92`), and therefore produces the correct prediction. We believe this concrete example makes the practical meaning of the reward-hacking phenomenon and the effect of SIRIUS much clearer.

---

> > ### Author Rebuttal · Reviewer_c2o4 · 2026-04-02
> >
> > The authors addressed my concerns, so I am maintaining my positive score.

---

### Official Review · Reviewer_XgKn · 2026-03-12

**Soundness:** 2
**Presentation:** 3
**Significance:** 2
**Originality:** 2
**Overall Recommendation:** 3
**Confidence:** 4

**Summary:**

This work studies reward hacking in LLM-based recommendation systems by investigating the negative sampling. The author provides empirical and theoretical analyses demonstrating that gradient updates on sampled pairs negligibly affect unsampled item rankings. Based on the analysis, the author further proposes SIRIUS to introduce a pseudo-negative item based on DPO and SimPO. The experimental results on three datasets show the effectiveness compared with three preference optimization methods.

**Compliance With Llm Reviewing Policy:**

Affirmed.

**Final Justification:**

The problem setup for a unique recommender system and the lack of baselines cannot persuade me to change the scores.

**Key Questions For Authors:**

1. How does the epsilon affect the theoretical analysis?

2. Besides introducing pseudo-negatives, some negative methods are also proposed(see weakness). I wonder if these methods also suffer from reward hacking?

3. How do the pseudo-negative behave if we scale the item sets?

**Limitations:**

Yes

**Strengths And Weaknesses:**

Strength:

1. The negative sampling in preference optimization has been vastly investigated in previous work. This work also addresses this topic and offers another perspective on reward hacking.

2. The work is organized well and easy to follow. The author first analyzes negative sampling for reward hacking and then provides a method guided by this principle.

3. The experiments are provided based on three datasets and provide baselines.

Weakness:

1. The related work and baselines are not so sound. We should discuss some work on negative sampling for preference optimization in LLM-based recommendation. For example, [1],[2],[3] .

2. The problem is not specific to LLM recommendation, but also some LLM human-value alignment. As a result, how about the research in the LLM field? And how about the method related to reward hacking?

3. The method heavily relies on some parameters. It seems somewhat arbitrary that the vanishing-gradient region accounts for such a large portion, as this behavior is largely dictated by the choice of epsilon.

4. The experiments should be enhanced by additional baselines, including both negative sampling and reward hacking methods.

[1] Ding C, Liu D, Wu J, et al. On Negative-aware Preference Optimization for Recommendation[J]. arXiv preprint arXiv:2508.09653, 2025.

[2] Wu J, Xie Y, Yang Z, et al. $\beta $-DPO: Direct Preference Optimization with Dynamic $\beta$[J]. Advances in Neural Information Processing Systems, 2024, 37: 129944-129966.

[3] Chen Y, Tan J, Zhang A, et al. On softmax direct preference optimization for recommendation[J]. Advances in Neural Information Processing Systems, 2024, 37: 27463-27489.

---

> ### Author Rebuttal · Authors · 2026-03-31
>
> Thank you for your thoughtful review and helpful suggestions.
>
> > Q1: The related work and baselines are not sufficiently sound. The paper should discuss more work on negative sampling and preference optimization in LLM-based recommendation, such as [1], [2], and [3].
> >
> > A1: We would like to clarify that SDPO [3] is already included as a baseline in our paper. It appears not only in the appendix, but also in the main text, including the introductory teaser figure and the main experimental results, where we report both its recommendation performance and its reward-hacking behavior. We also analyze in the appendix that SDPO can mitigate reward hacking to some extent. For [1], we agree that it is relevant as a negative-sampling-based preference optimization method. However, it is complementary rather than contradictory to our method, since SIRIUS can in principle be combined with such sampling strategies. For [2], its main focus is different from directly mitigating the type of reward hacking studied in our paper, so we do not view it as a directly comparable reward-hacking baseline. We will revise the related-work section to make these distinctions clearer.
>
> > Q2: The problem is not specific to LLM recommendation, but also appears in the broader LLM literature. How does this relate to reward-hacking research in that field?
> >
> > A2: We agree that reward hacking is not unique to recommendation, and that broader LLM training settings have also revealed related issues. Our view is that different reward-hacking mitigation methods often target different failure modes. For example, some methods are motivated by data-dependent patterns, such as POWER-DL, while others are designed to reduce specific biases such as length-related effects, e.g., SimPO with length normalization and ODIN through orthogonal decoupling of the length-related loss. In contrast, our paper focuses on a different mechanism: weak preference propagation in DPO/SimPO-style optimization under a large candidate space. We will strengthen the discussion to better position our work relative to broader LLM reward-hacking research while clarifying the difference in the failure mode addressed here.
>
> > Q3: The method seems to rely heavily on the choice of ε. How does ε affect the theoretical analysis?
> >
> > A3: We agree that ε is a user-chosen tolerance parameter, and the size of the ε-insensitive region naturally depends on this choice. Our intention is not to claim that one single ε value alone establishes the phenomenon. Rather, ε operationalizes how small an influence is considered practically negligible. In the paper, we provide the full distributional statistics instead of relying only on one threshold, and we use ε = 0.1 as a concrete and interpretable value to summarize the prevalence of the phenomenon. Therefore, the key point is not a single arbitrary threshold, but that the weak-propagation issue remains visible under a practically meaningful choice of ε.
>
> > Q4: Besides introducing pseudo-negatives, some negative-sampling methods have also been proposed. Do these methods also suffer from reward hacking?
> >
> > A4: We believe some of these methods can alleviate reward hacking to a certain extent, but they do not fundamentally remove the issue when the negative space is still very large. In fact, this is also consistent with our observations on SDPO [3]. As analyzed in our appendix and reflected in the main results, SDPO can reduce reward hacking compared with standard DPO, but the improvement is still limited because a small number of sampled negatives cannot fully cover the large set of potentially harmful unsampled negatives. This is exactly the motivation for introducing pseudo-negative anchors in SIRIUS.
>
> > Q5: How do the pseudo-negatives behave when the item or candidate set is scaled up?
> >
> > A5: We have reported additional results with different candidate-set sizes in the appendix, including settings such as 20, 50, and 100 candidates. As expected, the task becomes more challenging as the candidate set grows. However, under the same scaling trend, our method continues to show consistent relative improvements over the corresponding baselines. We report these results together with the reward-hacking perspective, rather than as a purely performance-oriented claim. We will make this point clearer in the revision.

---

> > ### Author Rebuttal · Reviewer_XgKn · 2026-04-01
> >
> > The discussion of limited baselines is insufficient for evaluating this work; the rebuttal has not fully addressed these concerns.

---

### Official Review · Reviewer_GNoj · 2026-03-13

**Soundness:** 3
**Presentation:** 3
**Significance:** 2
**Originality:** 2
**Overall Recommendation:** 3
**Confidence:** 4

**Summary:**

This paper attempts to address the reward hacking problem under a DPO-style LLM-based recommendation setting. It provides a \ epsilon region to measure how much optimizing between sampled pair (i,j) influence unsampled pair(i, k). The authors provide solutions SIRIUS by introducing a constant-reward pseudo item to mitigate the issue. The experiments demonstrate that \epsion-insensitive regions occur frequently in practice and that incorporating SIRIUS improves recommendation performance while mitigating reward hacking effects during training.

**Compliance With Llm Reviewing Policy:**

Affirmed.

**Key Questions For Authors:**

I do have a question about the basic settings on experiments. In the provided code, we see the raw item name making both historyList and itemList such as “311” and “Vast” in the steam dataset. This is more of a pattern-matching or classification problem, given the response is short(name only) and the candidate pool. I would like to see the performance of the experiment on SFT only first, compared with the later stage with RL methods, with different epochs along training.

Other questions, please see the weakness.

**Limitations:**

yes

**Strengths And Weaknesses:**

Strengths:
The statement in this paper is technically correct.
Presentation is clear and well-written.
Experiments are extensive and support its claisms.

Weaknesses:
The proposed theorem only depends on the logits and the output recommendation value,  and thus is general to all preference learning frameworks in recommendation. Do we observe a similar phenomenon in traditional recommendation which attempt to optimize pairwise scores? E.g., BPR[1] + matrix factorization is also under BT framework. If not, why is it unique to LLM-based recommendations?
I do not see a reasonable connection between thm3.5 and prop3.6. Basically, the enlargement of (i, j) could stem from either an increase in r_ii or a decrease in r_j. While we do not know which direction actually happens, the probability of either direction depends on the model parameters o. Intuitively, solely decreasing j would not affect p_{ik} at all, leading to an obvious existence of \epsilon region. The training dynamics of (p^{ij}(t), p^{ik}(t)) will not uniformly move in the 2-d space (p^{ij}, p^{ik}). However, \epsion region is quantified by its area in the (p^{ij}, p^{ik}) space while the training trajectories lie on a one-dimensional manifold determined by reward differences. Therefore, the area measure may significantly overestimate the practical impact of the insensitive region.

Given figure2 studies \epsion region of (p^{ij}, p^{ij}), the reason it plots p^{kj} vs p^{ij} is not clear.
This paper practically introduced r0 with a huge reward (e.g., even larger than r_i) in SimPO to affect p^{ik} vs p^{i0}, while leaving  p^{ik} vs p^{ij} unchanged. Given it is an effective way to increase the gap between r_i and r_k, it remains to be discussed whether a generic way of increasing r_i would work. E,g, does changing the reward of r0 to an arbitrary positive value still hold the result?

[1] Rendle, Steffen, et al. "BPR: Bayesian personalized ranking from implicit feedback."

---

> ### Author Rebuttal · Authors · 2026-03-31
>
> Thank you for your careful review and detailed comments. We apologize that some parts of our presentation, especially the relation between the theoretical results and the figures, were not sufficiently clear.
>
> > Q1: The proposed theorem seems general to pairwise preference learning. Does a similar phenomenon also appear in traditional recommendation methods such as BPR + MF, and if so, why is it discussed here in the context of LLM-based recommendation?
> >
> > A1: Our intention is not to claim that this phenomenon is unique to LLM-based recommendation. Rather, our goal is to systematically analyze and mitigate it in the DPO-style LLM recommendation setting studied in this paper. More specifically, Theorem 3.5 is intended to show the structural existence of an ε-insensitive region for general pairwise preference models under the assumptions in our analysis, while our experiments and method focus on the practical setting of LLM-based recommendation. Whether the same phenomenon also appears in traditional pairwise recommendation methods such as BPR + MF is an interesting question, but it is beyond the scope of the current paper and could be investigated in future work.
>
> > Q2: What is the exact relation between Theorem 3.5 and Proposition 3.6? Also, does the area of the ε-insensitive region overestimate its practical impact, since training trajectories do not uniformly move in the 2-D probability space?
> >
> > A2: Theorem 3.5 and Proposition 3.6 serve different roles. Theorem 3.5 establishes the existence of an ε-insensitive region in a general pairwise preference setting, while Proposition 3.6 specializes this result to the Bradley-Terry model and quantitatively analyzes the corresponding area. We agree that the area is not a complete description of the actual training dynamics, and should be viewed only as a geometric proxy rather than a full dynamical measure. To complement this theoretical quantity, we also provide an empirical statistic in Section 5.1 and Figure 3(a): when ε = 0.1, about 40% of samples in practice fall into the ε-insensitive region. Therefore, our claim does not rely on the area measure alone; the empirical evidence further supports the practical relevance of the phenomenon.
>
> > Q3: Figure 2 is unclear. If the ε-insensitive region is defined for the influence of a trained pair on an untrained pair, why are the plotted axes chosen in this way?
> >
> > A3: We apologize that Figure 2 was not sufficiently clear. Our definition of the ε-insensitive region is not about a single pair in isolation; it is specifically about how optimizing a sampled training pair `(i, j)` influences an unsampled pair `(i, k)`. In other words, the quantity of interest is the propagation from the trained preference relation to an unobserved one. The axes in Figure 2 are chosen to visualize exactly this influence relation, rather than to characterize a standalone property of `p^{ij}` itself. We will revise the caption and the corresponding explanation in the main text to make this point explicit.
>
> > Q4: Does SIRIUS rely on introducing a very large `r_0`? Would a more generic way of increasing the reward gap also work?
> >
> > A4: Please see our response to Reviewer mfzh, Q2.
>
> > Q5: The experimental setup seems close to pattern matching or classification in the Steam dataset. Why not focus more on SFT-only performance first?
> >
> > A5: The primary goal of this paper is not to propose a method that only improves final recommendation performance in an absolute sense, but to analyze and mitigate reward hacking during preference optimization. For this reason, our main experimental focus is the training dynamics, namely whether reward improvement becomes decoupled from ranking quality as training proceeds, and whether SIRIUS can mitigate this effect. While some item names in the dataset are short, the key question in our setting is still whether preference optimization can improve global ranking over the candidate set rather than merely improve the sampled pairwise objective. The ranking results are therefore presented together with the reward-hacking dynamics because the latter is the central phenomenon studied in this work.

---

> > ### Author Rebuttal · Reviewer_GNoj · 2026-04-04
> >
> > I still have the following questions. First, I still do not see the reason why this phenomenon is “unique” to DPO-style recommender system where the recommender is based on BT model. Second, My main concern is the $\epsilon$ area is not a proper measure since it obviously “over-estimates” the problem without analysis of training dynamis. However, author stil does not sufficiently explain this. Authors also argue that the empirical study is on preference learning, nevertheless, if there is no evidence showing the any superior over SFT, I believe, any contribution in further reinforcement learning stage will be discounted.

---

### Official Review · Reviewer_mfzh · 2026-03-13

**Soundness:** 3
**Presentation:** 2
**Significance:** 3
**Originality:** 3
**Overall Recommendation:** 4
**Confidence:** 4

**Summary:**

This paper mainly focuses on the reward hacking problem in existing LLM-based recommendation methods. The authors provide a theoretical analysis of this issue from the perspective of gradient propagation, and they innovatively introduce the concept of the "ε-insensitive region". They argue that the "ε-insensitive region" is not a flaw of any specific optimization algorithm, but rather an inherent structural property of pairwise preference models. Furthermore, using the Bradley–Terry model as a concrete case study, the authors show that the "ε-insensitive region" is not merely a theoretical possibility, but can occupy a substantial portion of the preference space, thereby hindering performance improvement in practical applications. To address this issue, the authors propose SIRIUS, a method that uses pseudo-negative samples for simulated preference optimization to mitigate reward hacking. The authors conduct experiments on three public benchmark datasets and find that a large proportion of data points lie within the "ε-insensitive region", suggesting that this phenomenon is fairly prevalent. Extensive experiments further show that SIRIUS can consistently mitigate reward hacking and improve ranking accuracy.

**Compliance With Llm Reviewing Policy:**

Affirmed.

**Key Questions For Authors:**

1. The analysis of the "ε-insensitive region" in this paper is mainly based on the Bradley–Terry model. Could the authors further discuss whether this phenomenon still exists under more complex reward models or more general preference learning frameworks?
2. Could the authors provide more experimental analysis on pseudo-negatives? For example, does the construction of pseudo-negatives significantly affect training stability? Is the model performance sensitive to the specific choice of pseudo-negatives?
3. If some reward hacking mitigation methods can be transferred to the current setting, could the authors include broader comparison experiments? If not, could the authors more clearly explain the differences between these methods and the proposed approach?

**Limitations:**

1. The theoretical analysis is based on the strong modeling assumption of the Bradley–Terry model, and its generalizability still requires further verification.
2. The design space of pseudo-negatives remains large, and the paper does not systematically analyze different design strategies.

**Strengths And Weaknesses:**

Strengths:
1. The paper provides a theoretical analysis of the reward hacking problem in DPO-style preference optimization methods for recommender systems, and explains the mechanism behind the occurrence of reward hacking.
2. The authors introduce the concept of the "ε-insensitive region" and derive an analytical expression for preference propagation under the Bradley–Terry model, theoretically explaining why updates on training pairs have only weak influence on the ranking of unsampled pairs.
3. The authors propose a new method for mitigating reward hacking, which does not require complex modifications to the model architecture and is highly compatible with existing training strategies.
4. The authors conduct extensive experiments on multiple public benchmarks, which provide strong validation for their theoretical findings and demonstrate the effectiveness of SIRIUS.

Weaknesses:
1. The quantitative analysis of the "ε-insensitive region" is mainly built upon the Bradley–Terry preference model, whereas the reward structure in real LLM-based recommender systems is often much more complex. The paper provides limited discussion on whether the theoretical results generalize to more general models.
2. The core idea of SIRIUS is to introduce pseudo-negatives, but the paper does not provide sufficient analysis of their generation strategy, distributional properties, or their impact on training stability.
3. The paper mainly compares with DPO and its variants, but lacks systematic comparisons with a broader range of reward hacking mitigation methods. If some existing reward hacking mitigation methods can be adapted to the current recommendation setting, comparisons with representative methods should be included; if such methods cannot be directly adapted, the discussion section should more clearly explain the connections and differences between this work and related approaches, such as ODIN, InfoRM, POWER / POWER-DL, and EPPO.

---

> ### Author Rebuttal · Authors · 2026-03-31
>
> Thank you for your thoughtful review.
>
> > Q1: The quantitative analysis of the ε-insensitive region is mainly based on the Bradley-Terry model. Does this phenomenon still exist under more complex reward models or more general preference-learning frameworks?
> >
> > A1: Our analysis contains both a qualitative and a quantitative part. The qualitative analysis is not restricted to the Bradley-Terry model; it applies more generally to pairwise preference models, which is also what our theoretical proof aims to show. We use Bradley-Terry for the quantitative analysis because it allows a tractable characterization of the ε-insensitive region under a shared formulation. For more complex reward models, we believe the phenomenon still exists qualitatively, while the exact proportion of the insensitive region and the corresponding choice of pseudo-negatives depend on the specific model.
>
> > Q2: The paper does not provide sufficient analysis of pseudo-negative generation, distributional properties, or training stability. Is the method sensitive to the specific choice of pseudo-negatives?
> >
> > A2: The purpose of pseudo-negatives is to move harmful samples away from the ε-insensitive region while avoiding training instability. We additionally tested nearby choices in the SimPO setting and observed that the overall behavior is not highly sensitive around our chosen value. On LastFM, the corresponding `HitRatio@1` trajectories are:
> >
> > | `r` | epoch 0 | epoch 1 | epoch 2 | epoch 3 | epoch 4 | epoch 5 |
> > | ----- | ------- | ------- | ------- | ------- | ------- | ------- |
> > | 0     | 0.624   | 0.659   | 0.697   | 0.724   | 0.730   | 0.732   |
> > | 1     | 0.624   | 0.658   | 0.695   | 0.717   | 0.731   | 0.726   |
> > | 10    | 0.624   | 0.664   | 0.694   | 0.723   | 0.732   | 0.725   |
> >
> > These results suggest that, in the SimPO setting and within this range, performance is not highly sensitive to the exact pseudo-negative value. We have not exhaustively explored more aggressive pseudo-negative constructions. For DPO, the choice is more consequential because the reward is unbounded, so the pseudo-negative value directly affects how many harmful samples can be pushed out of the ε-insensitive region.
>
> > Q3: The paper mainly compares with DPO and its variants, but lacks broader comparisons with reward-hacking mitigation methods. Could such methods be transferred to this setting, and if not, what are the differences from SIRIUS?
> >
> > A3: We agree that broader positioning would strengthen the paper. In our view, these methods fall into two cases. ODIN and POWER-DL are closer to our setting and can, in principle, be transferred, but their practical value is limited for different reasons. ODIN is designed to decouple a specific spurious reward component from the target signal. This is conceptually related to our goal, but in recommendation such decoupling would require much richer information over many negatives, which becomes costly when the item space is very large. POWER-DL is also transferable in principle, but its motivation is tied to reward-hacking patterns induced by particular data types in its original setting. In recommendation, the corresponding patterns do not appear to dominate the data distribution to the same extent. We also conducted experiments with POWER-DL on LastFM using Llama2, and observed that the reward margin keeps increasing while `HitRatio@1` quickly saturates, indicating that POWER-DL still suffers from reward hacking in DPO-based recommendation:
> >
> > | Epoch         | 1      | 2      | 3      | 4      | 5      |
> > | ------------- | ------ | ------ | ------ | ------ | ------ |
> > | Reward Margin | 18.05  | 19.36  | 23.39  | 25.69  | 26.84  |
> > | HitRatio@1    | 0.6060 | 0.6240 | 0.6333 | 0.6289 | 0.6297 |
> >
> > By contrast, InfoRM and EPPO are developed for optimization settings closer to explicit reinforcement learning, whereas our paper studies reward hacking in DPO/SimPO-style preference optimization with implicit rewards. We therefore do not view them as directly comparable baselines in the current setting.
>
> > Q4: The design space of pseudo-negatives remains large, and the paper does not systematically analyze different design strategies.
> >
> > A4: We agree that the design space of pseudo-negatives is broad. In this work, we focus on a simple design that provides a reasonable trade-off in both DPO and SimPO without additional computational overhead, while a more systematic exploration is left for future work.

---

> > ### Author Rebuttal · Reviewer_mfzh · 2026-04-03
> >
> > Some concerns are not fully addressed, and I will keep my score unchanged for now.
> > 1. Q2's response only shows limited robustness for a few SimPO values, but does not clarify the design principle, distributional role, or whether pseudo-negatives are truly necessary beyond simply enlarging the reward gap.
> > 2. Q2's current results are based on a small sensitivity check on LastFM with HR@1 only, and do not yet address DPO, the reward-hacking dynamics themselves, or training stability in a more direct way.
> > 3. Q3's response is directionally reasonable, but still somewhat incomplete: it gives plausible arguments for why some prior methods may not transfer cleanly, yet does not fully establish what is uniquely suited to SIRIUS in this setting.

---

### Official Review · Reviewer_cfsG · 2026-03-13

**Soundness:** 3
**Presentation:** 3
**Significance:** 3
**Originality:** 2
**Overall Recommendation:** 4
**Confidence:** 3

**Summary:**

This paper investigates the phenomenon of reward hacking in LLM-based recommendation systems, where training reward margins inflate while actual ranking performance stops improving. To mitigate this, the authors propose SIRIUS, which introduces pseudo-negative virtual anchors to provide non-vanishing gradients across the item universe.

**Compliance With Llm Reviewing Policy:**

Affirmed.

**Final Justification:**

The rebuttal clarified on the questions. I keep my original score.

**Key Questions For Authors:**

Does SIRIUS eventually suffer from reward hacking if trained for multiple epochs (for example, 20+), or does the pseudo-negative anchor provide a lower bound for ranking performance?

**Limitations:**

yes

**Strengths And Weaknesses:**

Strengths
1. The paper provides a formal definition and proof for the existence of $\epsilon$-insensitive regions, moving beyond heuristic explanations of reward hacking.
2. The use of pseudo-negative virtual anchors is a clever, computationally efficient alternative to exhaustive negative sampling.
3. The framework (SIRIUS) is tested across three benchmarks, compared with  multiple baselines, and shows consistent performance improvements


Weakness
1. The traditional sequential recommendation baselines (GRU4Rec, Caser, SASRec) are somewhat dated. Comparison against more recent non-LLM sequential models (e.g., GNN-based or Transformer-based models, such as “GRACE: Generative Recommendation via Journey-Aware Sparse Attention on Chain-of-Thought Tokenization”) would strengthen the claim of genuine recommendation gains.

---

> ### Author Rebuttal · Authors · 2026-03-31
>
> Thank you for your helpful comments and suggestions.
>
> > Q1: The traditional sequential recommendation baselines are somewhat dated, and comparison with more recent non-LLM sequential models would strengthen the claim of genuine recommendation gains.
> >
> > A1: We agree that including more recent non-LLM sequential recommendation baselines would further strengthen the empirical evaluation. In the current version, our primary focus is on the LLM-based recommendation setting, especially the comparison with existing LLM-based recommendation baselines in terms of reward-hacking behavior and its impact on ranking quality. For this reason, we included classical non-LLM sequential models such as GRU4Rec, Caser, and SASRec as representative references. We agree that this part can be further strengthened, and we will consider including more recent non-LLM sequential recommenders in a future version.
>
> > Q2: Does SIRIUS eventually suffer from reward hacking if trained for multiple epochs (e.g., 20+), or does the pseudo-negative anchor provide a lower bound on ranking performance?
> >
> > A2: We have not yet included additional experiments with substantially longer training horizons such as 20+ epochs, mainly because of the computational cost. Therefore, we do not claim that the pseudo-negative anchor provides a strict lower bound on ranking performance, since the eventual behavior can vary with the data distribution and training dynamics. Instead, what our current theory and experiments support is a relative claim: compared with the baselines, SIRIUS substantially mitigates reward hacking.

---

> > ### Author Rebuttal · Reviewer_cfsG · 2026-04-04
> >
> > Thanks for the explanations. I keep my original positive score.

---

### Decision · Program_Chairs · 2026-04-30

**Decision:**

Accept (regular)

**Comment:**

This paper studies reward hacking in LLM-based recommendation under preference optimization and proposes SIRIUS, a pseudo-negative based mitigation strategy motivated by a theoretical analysis of so-called epsilon-insensitive regions. Reviewers broadly agreed that the problem is timely and meaningful, and several found the theoretical framing useful for explaining why pairwise preference improvements may fail to translate into better global ranking behavior. The proposed mitigation is simple, compatible with existing optimization pipelines, and empirically supported across multiple benchmarks.

The main concerns were about scope and positioning rather than about a clear technical flaw. In particular, reviewers questioned whether the phenomenon is truly specific to LLM-based recommendation, whether the baseline set is broad enough, and whether the pseudo-negative mechanism and its stability properties are analyzed in sufficient depth. Two reviewers remained negative after rebuttal, largely because they still found the broader comparison and domain-specific framing incomplete.

At the same time, the rebuttal clarified several important points, including the relation to existing baselines and the intended scope of the paper's claims. My reading is that the remaining weaknesses concern breadth of comparison and positioning more than validity of the core contribution. Given the importance of the problem, the coherent theory-to-method pipeline, and the overall positive review balance, I support a Weak accept decision.